# Amyloid modifier SERF1a interacts with polyQ-expanded huntingtin-exon 1 via helical interactions and exacerbates polyQ-induced toxicity

Tien-Ying Tsai[1,2,3,9], Chun-Yu Chen[1,9], Tien-Wei Lin[1], Tien-Chang Lin[4,5], Feng-Lan Chiu[6], Orion Shih[4], Ming-Yun Chang[1,7], Yu-Chun Lin[1], An-Chung Su[5], Chiung-Mei Chen[8], U-Ser Jeng[4,5], Hung-Chih Kuo[6], Chi-Fon Chang[1] & Yun-Ru Chen [1✉]

Abnormal polyglutamine (polyQ) expansion and fibrillization occur in Huntington's disease (HD). Amyloid modifier SERF enhances amyloid formation, but the underlying mechanism is not revealed. Here, the fibrillization and toxicity effect of SERF1a on Htt-exon1 are examined. SERF1a enhances the fibrillization of and interacts with mutant thioredoxin (Trx)-fused Httex1. NMR studies with Htt peptides show that TrxHttex1-39Q interacts with the helical regions in SERF1a and SERF1a preferentially interacts with the N-terminal 17 residues of Htt. Time-course analysis shows that SERF1a induces mutant TrxHttex1 to a single conformation enriched of β-sheet. Co-expression of SERF1a and Httex1-polyQ in neuroblastoma and lentiviral infection of SERF1a in HD-induced polypotent stem cell (iPSC)-derived neurons demonstrates the detrimental effect of SERF1a in HD. Higher level of SERF1a transcript or protein is detected in HD iPSC, transgenic mice, and HD plasma. Overall, this study provides molecular mechanism for SERF1a and mutant Httex1 to facilitate therapeutic development for HD.

[1] Genomics Research Center, Academia Sinica, 128, Academia Rd., Sec. 2, Nankang District, Taipei 115, Taiwan. [2] Chemical Biology and Molecular Biophysics Program, Taiwan International Graduate Program, Institute of Biological Chemistry, Academia Sinica, 128, Academia Road, Sec. 2. Nankang, Taipei 115, Taiwan. [3] Institute of Biochemical Sciences, National Taiwan University, Taipei, Taiwan. [4] National Synchrotron Radiation Research Center, Hsinchu 300, Taiwan. [5] Department of Chemical Engineering, National Tsing Hua University, Hsinchu 300, Taiwan. [6] Institute of Cellular and Organismic Biology, Academia Sinica, 128, Academia Rd., Sec. 2, Nankang District, Taipei 115, Taiwan. [7] Taiwan International Graduate Program in Interdisciplinary Neuroscience, National Yang Ming Chiao Tung University and Academia Sinica, Taipei, Taiwan. [8] Department of Neurology, Linkou Chang Gung Memorial Hospital and College of Medicine, Chang Gung University, Taoyuan 333, Taiwan. [9] These authors contributed equally: Tien-Ying Tsai, Chun-Yu Chen. ✉email: yrchen@gate.sinica.edu.tw

Proteinaceous inclusions were found in many neurodegenerative diseases, including amyloid-β (Aβ) and tau in Alzheimer's disease (AD), α-synuclein in Parkinson's disease, and polyglutamine (polyQ) proteins in diseases attributed from expanded CAG trinucleotide repeats[1]. The most well-known polyQ disease is Huntington's disease (HD), an autosomal dominant fatal disorder caused by CAG expansion in the exon 1 of huntingtin (*HTT*) gene[2]. The expansion translates to mutant Htt proteins with an expanded polyQ tract flanked by N-terminal 17 amino acids (NT17) and C-terminal polyproline domain. Mutant Htts with more than 35–40 glutamine residues are prone to misfold and form amyloid fibrils that ultimately deposit as inclusions in the nucleus and cytoplasm of neurons, resulting in neuronal loss and progressive brain dysfunction[3–6]. The age onset of the disease is related to the length of the CAG trinucleotide expansion[2,7]. These patients have muscle coordination impairment, cognitive decline, and psychiatric disorder. The underlying pathogenic mechanism is currently under intensive investigation. However, effective therapy for HD is still lacking.

The Htt inclusions found in HD are accompanied by increase in oxidative stress and apoptosis[2,8]. Htt exon 1 can assemble into spherical oligomeric structures[9,10] which contribute to neuronal toxicity[11,12]. Another study using thioredoxin-polyQ fusion protein showed that the soluble β-sheet polyQ monomer is cytotoxic[13]. X-ray crystallography studies showed that Htt exon 1 with 17Q comprises a helical NT17, a flexible poly17Q region, and a helical polyproline region[14]. NMR studies showed that the NT17 domain and poly17Q track form partial helical structure[15,16]. Biochemical and simulation studies suggested that polyQ monomer with longer glutamine expansion forms heterogeneous collapsed structures in water[17]. PolyQ monomer is considered to be the critical nucleus for long polyQ fibrillization[18,19]. During fibrillization, soluble Htt oligomers appear in the early stage[10,20] and later, polyQ proceeds to form β-sheet fibrils with an interdigitated β-hairpin core[21].

PolyQ proteins and Q/N-rich prions are involved in complex structural transitions to form β-structures and generate neurotoxicity[22]. An essential role of coiled coil structure for aggregation was found in polyQ and Q/N-rich protein interactors, including Htt exon 1, and bioinformatics analysis showed that 63% Htt interactors contain or are predicted to contain coiled-coil structures[23]. Coiled coils are composed of multiple α-helical structures, in which the interfaces are based on mainly heptad repeat of residues. They play an essential role in protein–protein interactions, oligomerization, and other functions via the coiling of helices[24]. The coiled coil-destabilizing mutations of Htt were found to inhibit in vivo aggregation of Htt72Q and abolish cytotoxicity induced by Htt72Q[23]. Taken together, investigation of the expanded polyQ structure and the interactors involved in amyloid formation could facilitate the development of treatment for HD.

A novel gene called *modifier of aggregation 4* (*moag-4*, encoding protein MOAG-4) was identified in a chemical mutagenesis screen to modify protein aggregation[25]. Mutation or deletion of *moag-4* suppressed polyQ aggregation in polyQ *C. elegans* model, whereas transgenic overexpression of *moag-4* increased polyQ aggregation[25]. *MOAG-4* is an ortholog of human small EDRK-rich factor (*SERF*) 1a and *SERF2*. *SERF1a* is part of a 500 kb inverted duplication on chromosome 5q13 and highly related to spinal muscular atrophy. Human SERF1a protein has two isoforms. The short form contains 62 residues, and the long form has 110 amino acids. They share a highly conserved N-terminal region, and they are enriched in lysine and arginine. The amino acid sequence of SERF in various species is aligned and shown in Supplementary Fig. 1a. Both isoforms of human SERF1a are ubiquitously expressed throughout the central nervous system[26]. SERF1a was found to promote amyloid formation in vitro on several amyloid proteins, including α-synuclein, huntingtin, Aβ, and prion proteins[27]. SERF1a has been reported as an RNA-binding protein capable of mediating the functional integration of RNA in the nucleolus[28]. Given that SERF1a is a general modifier for amyloid formation, elucidating SERF1a structure and function is important for understanding neurodegenerative diseases. In the present study, we used short-form SERF1a as a model system to investigate the effect of SERF1a on Httex1-polyQ.

## Results

**Human SERF1a is a monomeric protein with high helical propensity that functions as an amyloid modifier.** We first analyzed the primary sequence of SERF1a and predicted its secondary structures using PHDsec algorithm (http://www.predictprotein.org/) (CUBIC, Columbia Univ, New York). The results showed that SERF1a was predicted to adopt a helix-loop-helix conformation containing 66% α-helices and 34% loop (Supplementary Fig. 1b). We constructed SERF1a with an N-terminal His-tag and thrombin cut site and overexpressed it in *E. coli*. SERF1a protein was purified to >95% purity (Supplementary Fig. 1c) after thrombin cleavage to remove the His-tag. Next, SERF1a was subjected to far-UV circular dichroism (CD) spectroscopy to examine the secondary structure (Supplementary Fig. 1d). The results showed a double minima spectrum that is consistent with previous literature[27,29]. Nuclear magnetic resonance (NMR) backbone assignment (NH, Cα, Cβ, and CO) for recombinant was 92% completed to further explore the structure details of SERF1a. A total of 54 out of 62 backbone amides were assigned under the experimental condition (pH 6.8 and 298 K, Supplementary Fig. 1e), while the unassigned residues included M1, K13, K47, Q48, K49, K54, K55, and Q58. The secondary structure elements of SERF1a were identified using CSI 3.0[30] on the basis of $^{13}C\alpha$, $^{13}C\beta$, $^{13}CO$, and $^{15}N$ chemical shifts of SERF1a (Supplementary Fig. 1f). The results indicated that SERF1a was composed of two helices, residues E8 to Q12 and A33 to A51, connected by a long loop. We added 10% trifluoroethanol (TFE), a helix-inducing agent[31,32], and found both helices were extended (R7-N14 and A33-K55). Thus, TFE treatment induced helical content by extending the length of helices. SERF1a was further subjected to sedimentation velocity examination in analytical ultracentrifugation (AUC) at 4 °C and 20 °C to examine its assembly (Supplementary Fig. 1g). The result showed only a single peak in C(s) distribution, in which the friction ratio of SERF1a was 1.68. These results demonstrated that SERF1a is predominantly monomeric with slightly extended conformation with α-helical structures.

To validate the amyloid effect of SERF1a, we examined the interaction with Aβ and α-synuclein by intrinsic fluorescence titration and its effect on Aβ and α-synuclein fibrillization by ThT assay. Fluorescence titration was performed by monitoring the tyrosine fluorescence of Aβ or α-synuclein titrated with SERF1a in different concentrations. The fluorescence was quenched upon the addition of SERF1a, indicating the interaction between SERF1a with Aβ and α-synuclein (Supplementary Fig. 2a). The quenching results were fitted, and the $K_D$ for Aβ and α-synuclein with SERF1a calculated were $24.9 \pm 9$ and $20.8 \pm 8.3 \mu M$, respectively. In the ThT assay, the addition of SERF1a accelerated both the fibrillization of Aβ and α-synuclein by enhancing the fibril elongation rates and the final ThT intensity (Supplementary Fig. 2b), where SERF1a also shortened the lag time of Aβ fibrillization, consistent with previous literature[27]. However, it should be aware that the amplitude of ThT signal cannot directly be considered as the amounts of the fibrils. The endpoint

products of Aβ and α-synuclein fibrillization with and without SERF1a were examined by transmission electron microscopy (TEM, Supplementary Fig. 2c). Similar fibril formation was found in Aβ and α-synuclein, in which the fibrils were more clustered in the presence of SERF1a. Together, the recombinant SERF1a was confirmed to be capable of enhancing amyloid fibrillization.

**Human SERF1a accelerates and enhances fibril formation of Htt exon 1 in a polyQ length-dependent manner.** Although SERF1a ortholog has been shown to modulate polyQ aggregation in *C. elegans*[25], the exact mechanism at the protein level has not been characterized. Htt-polyQ exon 1 proteins with different glutamine (Q) lengths with an N-terminal thioredoxin (Trx) fusion tag were constructed and purified to investigate whether SERF1a could enhance amyloid fibrillization. The Htt-polyQ exon 1 variants contained an N-terminal region with 17 amino acids, a polyQ region, and a proline-rich domain (PRD) at the C-terminus. The different glutamine (Q) lengths included normal Httex1, i.e., Httex1-15Q and Httex1-29Q, and mutant Httex1, i.e., Httex1-39Q and Httex1-49Q. First, the SERF1a effect on amyloid formation of various TrxHttex1-polyQ was examined by ThT assay. Data showed that SERF1a accelerated the fibrillization on mutant TrxHttex1-39Q and TrxHttex1-49Q, but it had little effect on normal TrxHttex1-15Q and TrxHttex1-29Q (Fig. 1a, Supplementary Data 1). The half-time of fibril elongation was at ~6 h for TrxHttex1-49Q in the presence of SERF1a. It was at ~12 h for TrxHttex1-49Q in the absence of SERF1a. The endpoint products were subjected to filter trap assay (Fig. 1b) and TEM imaging (Fig. 1c). The filter trap assay clearly showed an increase in aggregation for all TrxHttex1 variants in the presence of SERF1a. Even in normal TrxHttex1, SERF1a was able to increase the level of aggregates trapped on the filter member. TEM images showed that mutant TrxHttex1 formed long and mature amyloid fibrils in comparison to normal TrxHttex1, where fibril density and quantity further increased in the presence of SERF1a. Sparse and small aggregates found in TEM for TrxHttex1-15Q and TrxHttex1-29Q were also shown. Together, the results demonstrated that SERF1a enhances TrxHttex1 fibril formation in a polyQ length-dependent manner.

Next, time-course samples were collected during the aggregation for dot-blot and far-UV CD spectroscopy to examine conformational changes of TrxHttex1 in the absence or presence of SERF1a. Samples were collected at different incubation times, dotted, and subjected to dot blotting probed by 1C2 antibody specific for exposed polyQ domain[33] and MW7 antibody for the proline-rich domain[34] (Fig. 1d). The result showed that in the presence of SERF1a, the 1C2 antibody signal decreased faster than that without SERF1a. This effect was more profound in mutant TrxHttex1 than in normal TrxHttex1. By contrast, the MW7 signals did not change over time. As previous studies reported that 1C2 antibody specifically recognizes exposed, but not buried, expanded polyQ[33,35], the results of the present study indicated that SERF1a enhances the fibril formation that may lead to inaccessibility to polyQ expansion, especially in mutant TrxHttex1. The TrxHttex1-49Q time-course samples were also subjected to dot blot probed by amyloid conformation-specific antibodies, including A11, an antibody specific to amyloid prefibrillar oligomers[36] and OC, an antibody specific to fibril and fibrillar oligomers[37] (Supplementary Fig. 3). In the presence of SERF1a, immunoreactivity with A11 and OC antibodies increased after 12 h of incubation. In the absence of SERF1a, the signals appeared after 18 h. The incubation time for oligomers and fibrils detected by antibodies was correlated with the lag time observed in the ThT assay (Fig. 1a, Supplementary Data 1) and dot-blot assay (Fig. 1d). These data suggested that SERF1a also

accelerated and enhanced oligomer formation in mutant TrxHttex1-polyQ aggregation.

Changes in the time-dependent secondary structure in TrxHttex1-polyQ with and without SERF1a were monitored by far-UV CD spectroscopy to confirm the observations (Fig. 1e, Supplementary Data 1). The buffer background and SERF1a alone spectra were subtracted from TrxHttex1-polyQ spectra in the absence or presence of SERF1a, respectively. The results showed that in the absence of SERF1a, the normal and mutant TrxHttex1 adopted a mixed helical and random coil structure at time 0. After incubation, the mutant TrxHttex1 gradually formed β-sheet content with a single minimum at 216 nm. In the presence of SERF1a, it accelerated the β-sheet structure formation of mutant TrxHttex1 (TrxHttex1-39Q and TrxHttex1-49Q) but not to the same degree for normal TrxHttex1 (TrxHttex1-15Q and TrxHttex1-29Q, Fig. 1f, Supplementary Data 1). Although TrxHttex1-49Q, in the presence of SERF1a, showed lesser signal changes in 216 nm, the time to become β-structure occurred earlier at 3 h compared with that of TrxHttex1-39Q at 12 h. This result is consistent with the ThT assay and dot blot. We also examined the secondary structure of thioredoxin alone in the absence or presence of SERF1a by CD and found no effect of SERF1a on thioredoxin structure; therefore, we exclude the possibility that β-sheet structure formation might result from the effect of SERF1a on thioredoxin (Supplementary Fig. 4).

To investigate whether SERF1a integrates into or dissociates from Httex1 fibrils during the aggregation, we collected the end-point products from the aggregation assay and employed immunogold labeling to detect SERF1a in TrxHttex1-39Q fibrils (Supplementary Fig. 5a). The TEM results showed that there were very few immunogolds-conjugated antibody recognizing SERF1a in TrxHttex1-39Q and SERF1a co-incubated sample, which means that SERF1a dissociated from the fibrils without retention during the aggregation. No immunogold was detected in the TrxHttex1-39Q alone. This finding was also confirmed by western blot and SDS-PAGE (Supplementary Fig. 5b). TrxHttex1-39Q was incubated in the absence or presence of SERF1a to form the fibrils, and the samples were then centrifuged to separate the supernatant for soluble proteins and the pellet for insoluble fibrils. By probing with SERF1a antibody, we found that SERF1a could only be detected in the soluble proportion of the sample but not in the insoluble fraction, suggesting the absence of SERF1a in the TrxHttex1-39Q fibrils even if SERF1a was present during the aggregation process. SDS-PAGE result also showed that SERF1a was only present in the supernatant but not in the pellet. Together, these results revealed that the interaction of SERF1a interacts dynamically with TrxHttex1 and SERF1a does not remain associated with TrxHttex1-39Q at the fibril stage.

**SERF1a interacts with mutant Httex1, but not normal Httex1, and the interaction is enhanced by the level of helical content.** Although SERF1a enhances mutant Httex1 amyloid fibrillization and β-sheet transition, whether SERF1a interacts with the Htt-polyQ exon 1 protein is unknown. To examine the possible interaction between SERF1a and Httex1-polyQ, we used the intrinsic fluorescence of TrxHttex1-polyQ titrated with SERF1a to estimate the binding affinity. TrxHttex1-polyQ contained phenylalanines, the only aromatic residues, that were emitted at 282 nm, whereas SERF1a did not contain any aromatic residues contributing no fluorescence background. The intrinsic fluorescence of TrxHttex1 variants at 25 μM with various concentrations of SERF1a were collected and normalized to its signal without SERF1a (Fig. 2). The intrinsic fluorescence of mutant TrxHttex1 was quenched upon addition of SERF1a but normal TrxHttex1 was not, indicating that the interaction between TrxHttex1 and

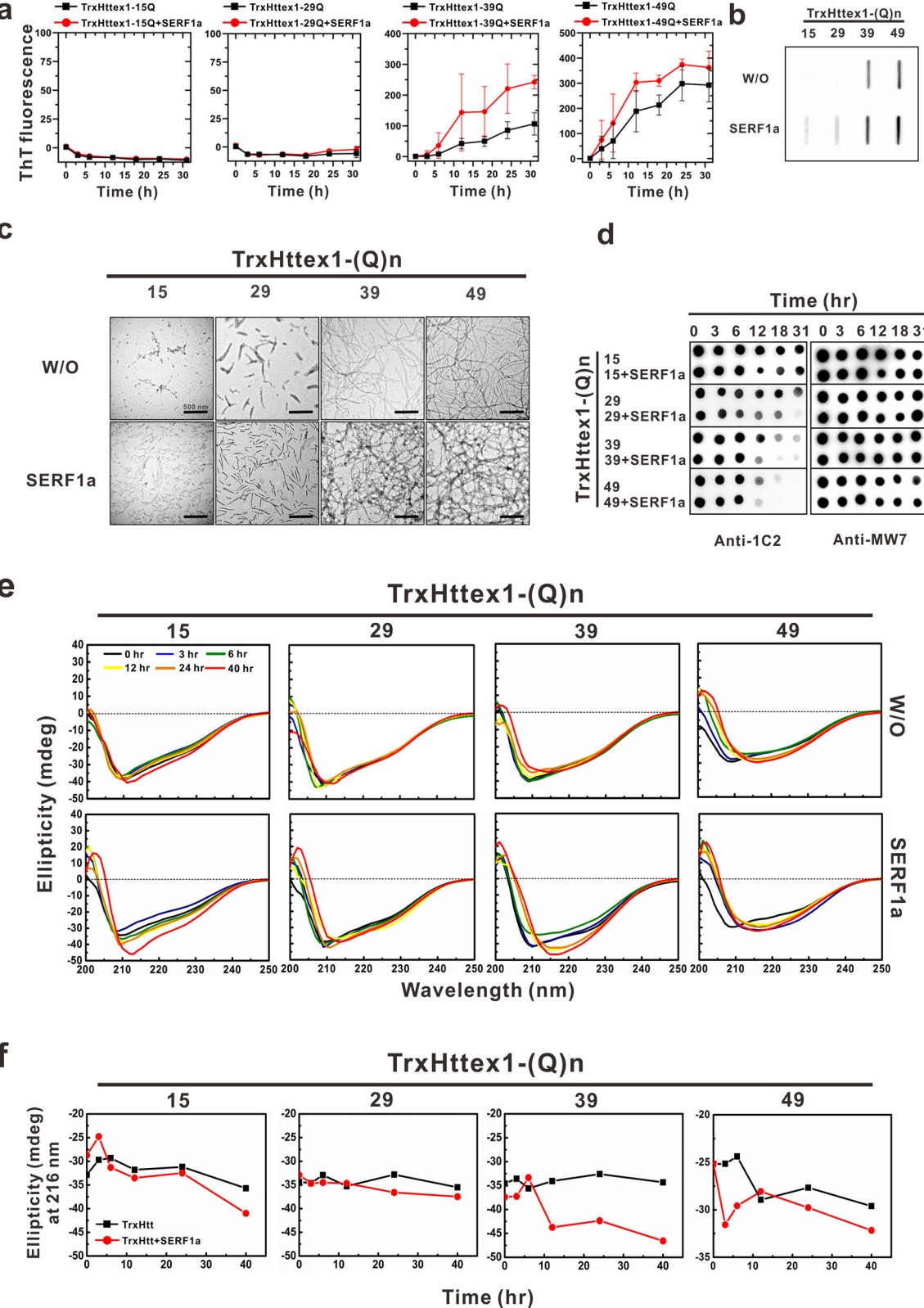

SERF1a is dependent on polyQ repeat-length. The results indicated that SERF1a predominantly interacted with mutant TrxHttex1 but not with normal TrxHttex1 (Fig. 2a, Supplementary Data 1). Whether thioredoxin fusion on Httex1 affects the result of SERF1a binding was also tested. Httex1-15Q and Httex1-39Q with thioredoxin removal were prepared and titrated with SERF1a in fluorescence titration assay. The fluorescence

quenching results showed that SERF1a only interacted with Httex1-39Q but not with Httex1-15Q (Fig. 2b, Supplementary Data 1). Therefore, thioredoxin fusion did not affect the interaction between SERF1a and Httex1.

PolyQ and polyQ interacting partners have been shown to interact via coiled coils[23]. Coiled coils are formed by two or more α-helical structures via homo- or hetero-complexes with

**Fig. 1 SERF1a enhances fibrillization of wild-type and mutant TrxHtt exon 1. a** ThT fluorescence assay of TrxHtt exon 1 in the absence or presence of SERF1a. Fifty μM of different polyQ-expanded Htt exon 1 proteins, including TrxHttex1-15Q, TrxHttex1-29Q, TrxHttex1-39Q, and TrxHttex1-49Q, were incubated at 37 °C with continuous shaking at 200 rpm with and without equimolar SERF1a. $n = 3$. The error bars represent standard deviation. **b** Filter retardation assay of the end-point products of ThT assay after incubation. More aggregates were found in TrxHttex1 with SERF1a. **c** TEM images of the end-products of ThT assay. The scale bars are 500 nm. **d** Dot blot of TrxHtt exon 1 variants with and without SERF1a at different incubation times. TrxHttex1 proteins were detected by 1C2 and MW7 antibodies. **e** Far-UV CD analysis of TrxHttex1 variants with and without SERF1a at different incubation times. Buffer background was subtracted from the spectra of TrxHttex1 alone, and SERF1a alone spectra were subtracted from that of TrxHttex1 with SERF1a. **f** The ellipticity at 216 nm from the far-UV spectra of panel (**e**) was plotted.

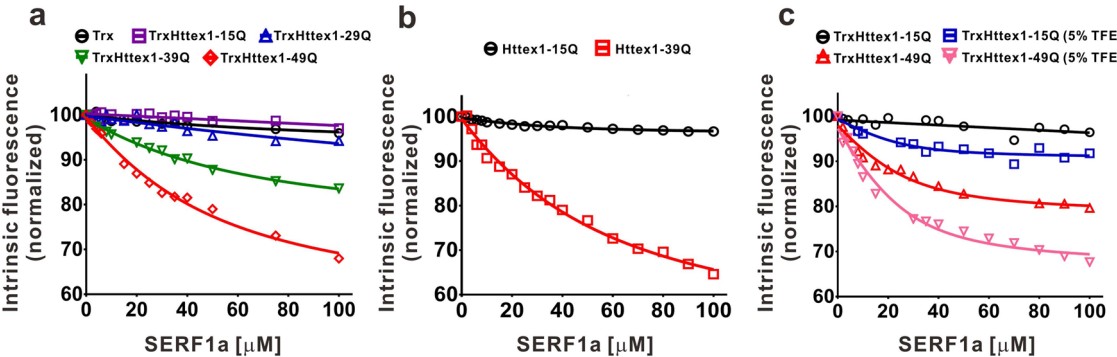

**Fig. 2 SERF1a interacts with Htt exon 1.** Htt exon 1 variants at 25 μM was titrated with SERF1a. Intrinsic fluorescence was monitored at an excitation of 257 nm and an emission of 282 nm. Signals were normalized to the starting signal. **a** Fluorescence quenching of TrxHttex1 variants upon titration with SERF1a. **b** Fluorescence quenching of Httex1-15Q and Httex1-39Q without thioredoxin tag upon titration with SERF1a. **c** Fluorescence quenching of TrxHttex1-15Q and TrxHttex1-49Q in the absence or presence of 5% TFE upon SERF1a titration.

conformal diversity[38]. The interactions can be presented by heptad repeats. In CD experiments, the $[\theta]222/[\theta]208$ ratio was used to evaluate the presence of coiled-coil helices[39,40], in which the ratio at 0.8–0.9 indicated non-associated α-helices and at $1.0 \pm 0.03$ for two-stranded coiled-coil helices. Far-UV CD spectroscopy was conducted to measure the α-helical content of SERF1a and TrxHttex1-polyQ variants in varying concentrations of TFE to examine whether helical content plays a role in Httex1 and SERF1a interaction, and the $[\theta]222/[\theta]208$ ratios were plotted against TFE concentration (Supplementary Fig. 6). The result showed that increasing TFE gradually promoted the α-helical content of SERF1a in and above 20% TFE (Supplementary Fig. 6a). TrxHttex1-39Q and TrxHttex1-49Q also increased α-helical content above 10% TFE, whereas TrxHttex1-15Q and TrxHttex1-29Q changed to a lesser extent (Supplementary Fig. 6b). Next, SERF1a and TrxHttex1-polyQ interaction with and without 5% TFE was further compared (Fig. 2c, Supplementary Data 1). In the presence of TFE, stronger fluorescence quenching was observed upon titration of SERF1a with TrxHttex1-15Q and TrxHttex1-49Q. The result indicated that the TFE-induced α-helical structures in SERF1a and TrxHttex1-polyQ enhanced the interaction. Thus, SERF1a and TrxHttex1-polyQ interaction could be enhanced upon the increase in helical content.

**Mutant Httex1 binds to helical regions of SERF1a.** To determine the Httex1-polyQ binding site on SERF1a, we performed NMR HSQC analysis of 50 μM $^{15}$N-labeled SERF1a with TrxHttex1-39Q titration from the molar ratio of SERF1a: TrxHttex1-39Q as 1:0.5 to 1:2. Upon increasing the concentration of TrxHttex1-39Q, the chemical shift perturbation (CSP) and intensity changes in $^{15}$N-SERF1a amide signals were detected (Fig. 3a). The TrxHttex1-39Q-induced CSP only observed for few residues A10, R11, I21, K23, and K27 (Fig. 3b, c, Supplementary Data 1) with minor shifts (> 0.1ppm), while the intensity changes in $^{15}$N-SERF1a amide signals could be detected clearly. The

intensity drop of each residue was normalized to K62 signal, and the residue showing the least intensity changes during titration was plotted (Fig. 3d and Supplementary Fig. 7, Supplementary Data 1). Almost all residues experienced intensity drop, and the residues showing intensity drop > 70% were G4, N5, Q6, A10, R11, N14, Q19, T32, A33, Q35, R36, Q38, and A50. This result indicated that the main regions of SERF1a affected by TrxHttex1-39Q addition were the N-terminus and α-helical regions (Fig. 3e).

**SERF1a primarily interacts with N-terminus of Httex1-polyQ peptides.** To examine possible SERF1a and Httex1 interaction via coiled-coils (CC), we designed several Htt peptides comprising 33 amino acids with substitution of residues in a/d positions in the helical wheel. The design and predicted coiled-coil position was based on previous literature[23] using the program DrawCoil[41]. The substitutions were either with leucine for CC-enhancing effect or proline for CC-destabilizing effect[23,42] (Fig. 4a). Five Htt peptides (Htt-1, -2, -3, -4, and -5) were generated on the basis of the hepated prediction of wild-type peptide (Htt-0). Htt-1 and Htt-2 were designed as four leucine residues replacing residues of position a/d in the polyQ region, while Htt-2 had additional three prolines replacing the position a/d in the N-terminal region. In Htt-3 and Htt-4, the position a/d in the polyQ region were substituted by four prolines, and Htt-4 included additional three prolines replacing leucines at the position a/d in the N-terminal region. Htt-5 had three prolines at the N-terminal a/d positions instead of leucine. In addition to the Htt peptides, NT17 containing N-terminal 17 amino acids of Htt was synthesized for the following examinations.

Far-UV CD spectra showed that the Htt-1 peptide with CC-enhancing effect in polyQ region displayed a classic α-helix structure (Fig. 4b, Supplementary Data 1). Htt-2 with CC-destabilizing effect in the N-terminus and CC-enhancing effect in polyQ region showed partial α-helix structure, which is similar to wild-type Htt peptide (Htt-0). However, the other three peptides, Htt-3 with CC-destabilized polyQ region, Htt-4 with all

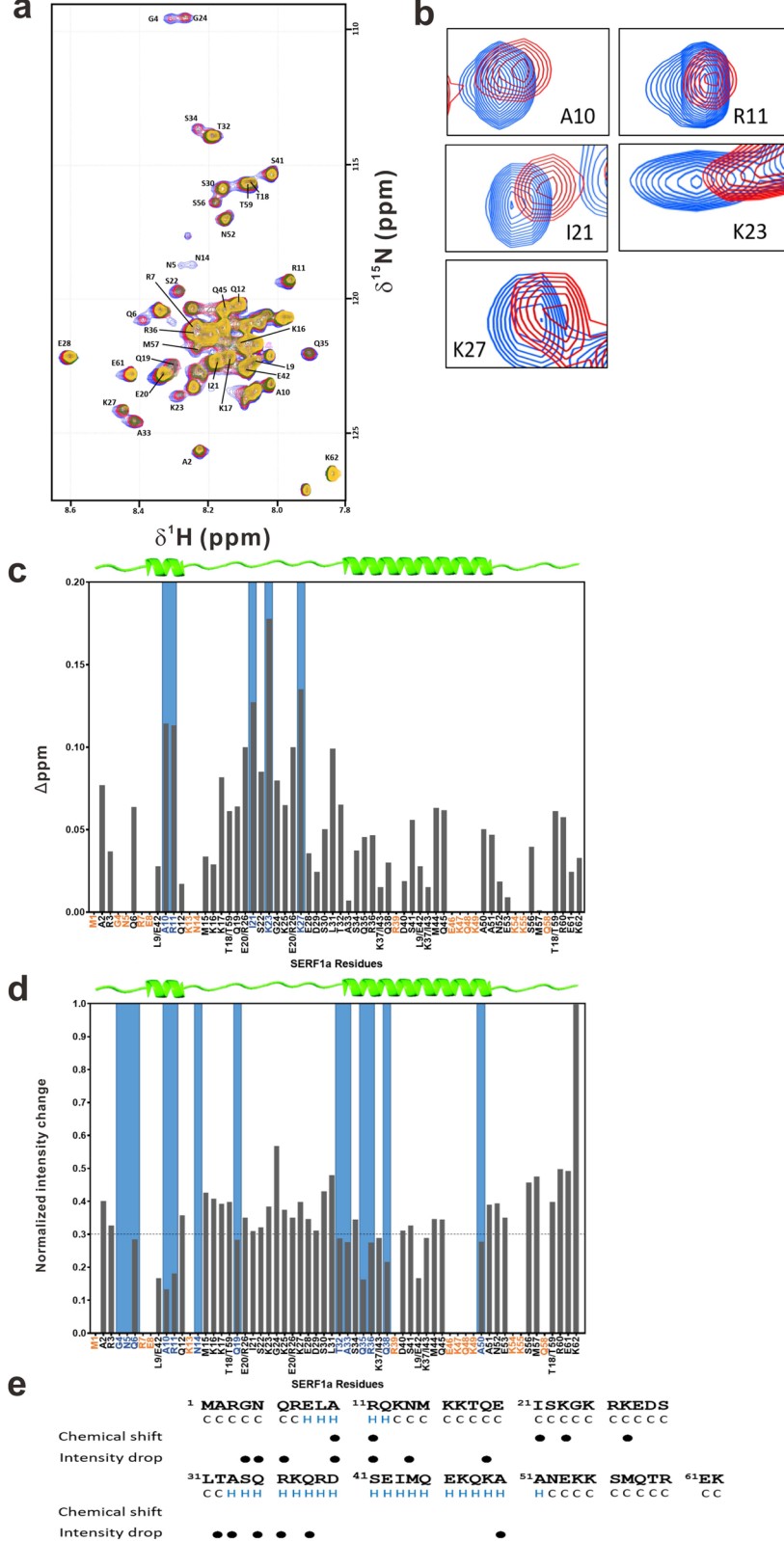

**Fig. 3 Interaction between TrxHttex1-39Q and SERF1a by NMR. a** Overlay of HSQC spectra of [15]N-labeled SERF1a with increasing ratios of TrxHttex1-39Q. **b** Residues with larger changes in chemical shift. Peaks in blue were SERF1a alone, and SERF1a with TrxHttex1-39Q at a ratio of 1:2 were labeled in red. **c** Chemical shift perturbation (CSP) and (**d**) intensity drop between control and SERF1a with TrxHttex1-39Q (1:2). The signals of residues L9/E42, T18/T59, and E20/R26 were overlapped and unable to be differentiated. Undetected residues were marked in orange. Residues with CSP > 0.1 ppm or residues with normalized intensity drop > 70% were highlighted in blue. **e** Secondary structure of SERF1a determined by chemical shift index (CSI 3.0). Residues with CSP > 0.1 ppm or residues with normalized intensity drop > 70% were marked with black dots.

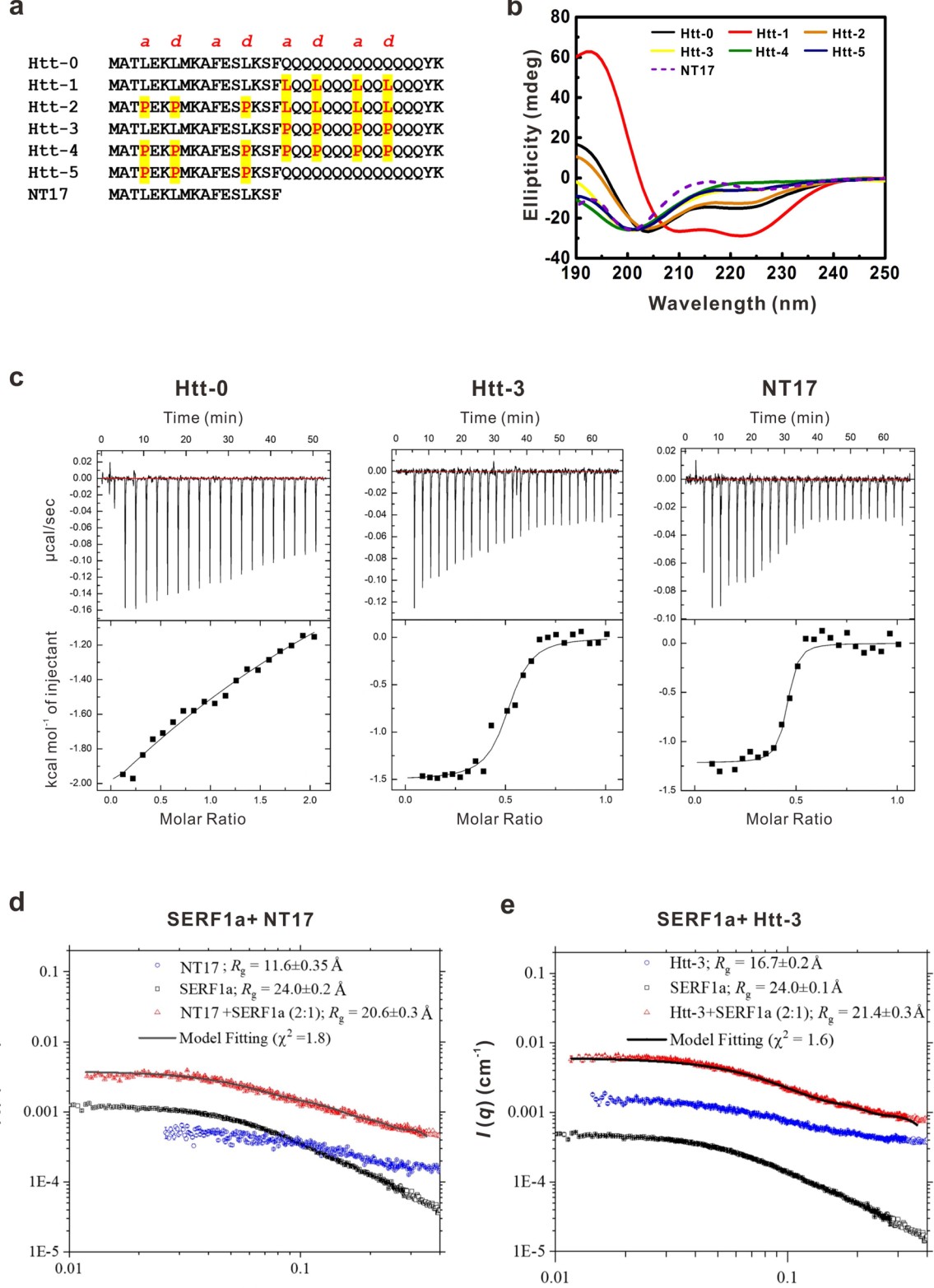

**Fig. 4 N-terminal Htt binds to SERF1a. a** Primary sequence of the designed Htt peptides. Point mutations of a/d positions in the heptads were highlighted in yellow. **b** Far-UV CD spectra of Htt peptides. **c** ITC data of Htt peptides with SERF1a. SERF1a interacts strongly with Htt-3 and NT17. **d** SAXS data and the extracted $R_g$ values of NT17, SERF1a, and the mixture in 2:1 molar ratio. Data for the mixture were fitted using a DAMMIN model. **e** SAXS data and $R_g$ values of Htt-3, SERF1a, and the mixture in 2:1 molar ratio. Data for the mixture were fitted using a DAMMIN model.

destabilized *a/d* positions, and Htt-5 with CC-destabilized N-terminus, all displayed random coil structure. NT17 displayed a partial α-helical structure consistent with previous studies[6,43]. The helical property of the Htt peptides in varying concentrations of TFE ranging from 0% to 20% was also measured by far-UV CD spectroscopy (Supplementary Fig. 8a). The result demonstrated that Htt-1 was fully helical regardless of TFE content, whereas Htt-4 was fully random coiled without the ability to form helices in 20% TFE. Htt-0, Htt-2, and NT17 adopted more similar helical transition, whereas Htt-3 and Htt-5 could be induced to form partial helices by TFE. The [θ]222/[θ]208 ratio is plotted in Supplementary Fig. 8b. By using sedimentation velocity (SV) experiment in analytical ultracentrifugation (AUC), we further examined the species of these Htt peptides and found that NT17, Htt-2, Htt-3, Htt-4, and Htt-5 were predominantly monomers, while Htt-0 was dimers and Htt-1 showed predominantly pentamers and octamers (Supplementary Fig. 9).

Next, isothermal titration calorimetry (ITC) was used to examine the binding affinity of SERF1a on these Htt peptides (Fig. 4c and Supplementary Fig. 10, Supplementary Data 1). SERF1a in different concentrations was titrated into Htt fragments. The ITC data were analyzed and fitted. The result showed that only NT17 and Htt-3 bind to SERF1a, where NT17 had the strongest interaction with SERF1a showing a $K_A$ value of $1.38 \times 10^7 \pm 6.01 \times 10^6 \, M^{-1}$ ($K_D = 7.25 \times 10^{-8}$ M, 0.0725 μM). The $K_A$ value of SERF1a and Htt-3 was $3.01 \times 10^6 \pm 9.67 \times 10^5 \, M^{-1}$ ($K_D = 3.32 \times 10^{-7}$ M, 0.33 μM). The binding stoichiometry for SERF1a to NT17 and Htt-3 ratio was close to 0.5, indicating one SERF protein binds to two Htt peptides. Other Htt fragments, including Htt-0, Htt-1, Htt-2, Htt-4, and Htt-5, did not bind to SERF1a. The result showed that the N-terminal fragment containing the first 17 amino acids (NT17) plays an important role in the interaction with SERF1a. Those peptides with N-terminal disruption, i.e., Htt-2, Htt-4, and Htt-5, were unable to interact with SERF1a. However, the wild-type Htt peptide, Htt-0, did not bind to SERF1a.

To further confirm the interaction between NT17 and SERF1a, we conducted small-angle x-ray scattering (SAXS) for NT17 and Htt-3 (Figs. 4d, e and Supplementary Fig. 11, Supplementary Data 1) and their mixture with SERF1a, respectively. The result showed that the absolute X-ray scattering intensity $I(0)$ of the NT17 and SERF1a co-incubated sample ($R_g = 20.6$ Å) was larger than the summation of the two respectively measured $I(0)$ values of NT17 monomer ($R_g = 11.6$ Å) and the SERF1a monomer ($R_g = 24.0$ Å), indicating the binding of NT17 and SERF1a (Fig. 4d, Supplementary Data 1) into a more massive complex for the higher $I(0)$ observed. We note that the SEC-SAXS method used allows us to verify that there is only one single species with one constant $R_g$ value observed for the mixture along the elution SAXS profiles (Supplementary Fig. 12). Similar behavior of the SAXS $I(0)$ values was also observed for Htt-3, SERF1a, and their mixture, suggesting complex formation of Htt-3 with SERF1a (Fig. 4e, Supplementary Data 1). These two results suggested that most likely, Htt-3 binds to SERF1a mainly via the NT17 domains, thus supporting the ITC results. Moreover, the concentration-independent data for SERF1a, NT17, Htt-3, and the mixture of SERF1a-NT17 and SERF1a-Htt-3 are shown (Supplementary Fig. 12), and all the concentration-independent profiles suggest that these species are monodisperse.

**SERF1a facilitates the mutant Httex1 to form a single soluble species.** To investigate the detailed function of SERF1a to promote Httex1 aggregation, we collected and analyzed TrxHttex1-49Q in the absence or presence of SERF1a at different incubation times, i.e., 0, 6, 12, 24, and 48 h, by SV-AUC and size-exclusion

chromatography (SEC) (Fig. 5 and Supplementary Figs. 13 and 14). In AUC, the samples were examined by monitoring the absorbance at 280 nm. The buffer control was run as reference and subtracted from TrxHttex1-49Q, while SERF1a did not contribute signal in TrxHttex1-49Q with SERF1a sample at 280 nm as it contained no aromatic residues. In the absence of SERF1a, the SV analysis showed TrxHttex1-49Q was predominantly monomeric with minor species of possibly trimer and tetramer. The sedimentation coefficient decreased from 1.72 S to 1.56 S during the incubation. In the presence of SERF1a, the TrxHttex1-49Q mainly showed a single species distribution in solution with very few trimers, and the sedimentation coefficient remained at ~1.54 S (Fig. 5a and Supplementary Fig. 13, Supplementary Data 1).

Next, time-course samples of TrxHttex1 with and without SERF1a were subjected to SEC analysis (Fig. 5b and Supplementary Fig. 14, Supplementary Data 1). The samples were collected and centrifuged, and the supernatant was injected to SEC monitored at an absorbance of 280 nm. Considering that SERF1a contained no aromatic residues, the 280 nm signal represented soluble TrxHttex1 protein, where the insoluble TrxHttex1 was pelleted. In the SEC chromatogram, at time 0, the result showed TrxHttex1-49Q alone was eluted in a wide peak from fraction 11 to 14 and two small peaks consisting of fractions 15 and 16 and fraction 17. During incubation, fractions 11–14 gradually decreased, whereas the other two fractions gradually increased. In the presence of SERF1a, the three peaks remained, in which fractions 11–14 and fraction 17 were the two major peaks. During incubation, fraction 17 increased, whereas other fractions were not observed. After 48 h of incubation, the peaks of TrxHttex1-49Q alone showed more protein in fractions 15 and 16 and fraction 17, while TrxHttex1-49Q with SERF1a predominantly showed a single peak on fraction 17. The time-course samples for fractions 14 and 17 were further collected and subjected to native PAGE (Fig. 5c). In the absence of SERF1a, fractions 14 and 17 migrated differently, in which fraction 17 migrated faster and had three bands. The difference in protein migration found in native-PAGE indicated the conformational differences residing in TrxHttex1-49Q. In the presence of SERF1a, fraction 14 disappeared after incubation, consistent with the SEC result. Interestingly, fraction 17 became majorly one band. This result is consistent with the AUC analysis showing that SERF1a accelerated the transformation of different species of TrxHttex1-49Q to a more defined conformation. To further examine the conformation of Httex1 with the presence of SERF1a, we incubated TrxHttex1-49Q with SERF1a for 48 h and loaded the sample onto SEC to isolate fraction 14 and 17. The collected samples were subjected to dot blot (Fig. 5d) and far-UV CD measurement (Fig. 5e, Supplementary Data 1). The result showed that SERF1a was only presented in fraction 17, but not in fraction 14. The CD spectra of fraction 17 were subtracted to that of SERF1a alone and the result showed TrxHttex1 formed a β-sheet conformation with a minimal around ~216 nm (Fig. 5e, Supplementary Data 1), whereas fraction 14 is like α-helical structure showing two double minimum. Together, the results suggested that SERF1a accelerates conformational transition of monomeric TrxHttex1-49Q to a β-conformer, that is consistent with the previously reported toxic β-sheet Httex1 monomer[13].

**SERF1a promotes Httex1 aggregation and induces Httex1 toxicity in neuroblastoma.** To determine whether the aggregation and toxicity of Htt exon 1 arise from SERF1a in cells, we co-expressed EGFP-tagged Htt exon 1 with 25Q (EGFP-Httex1-25Q) or 109Q (EGFP-Httex1-109Q) and SERF1a-myc or myc-only plasmids in mouse neuroblastoma Neuro-2A and subjected to

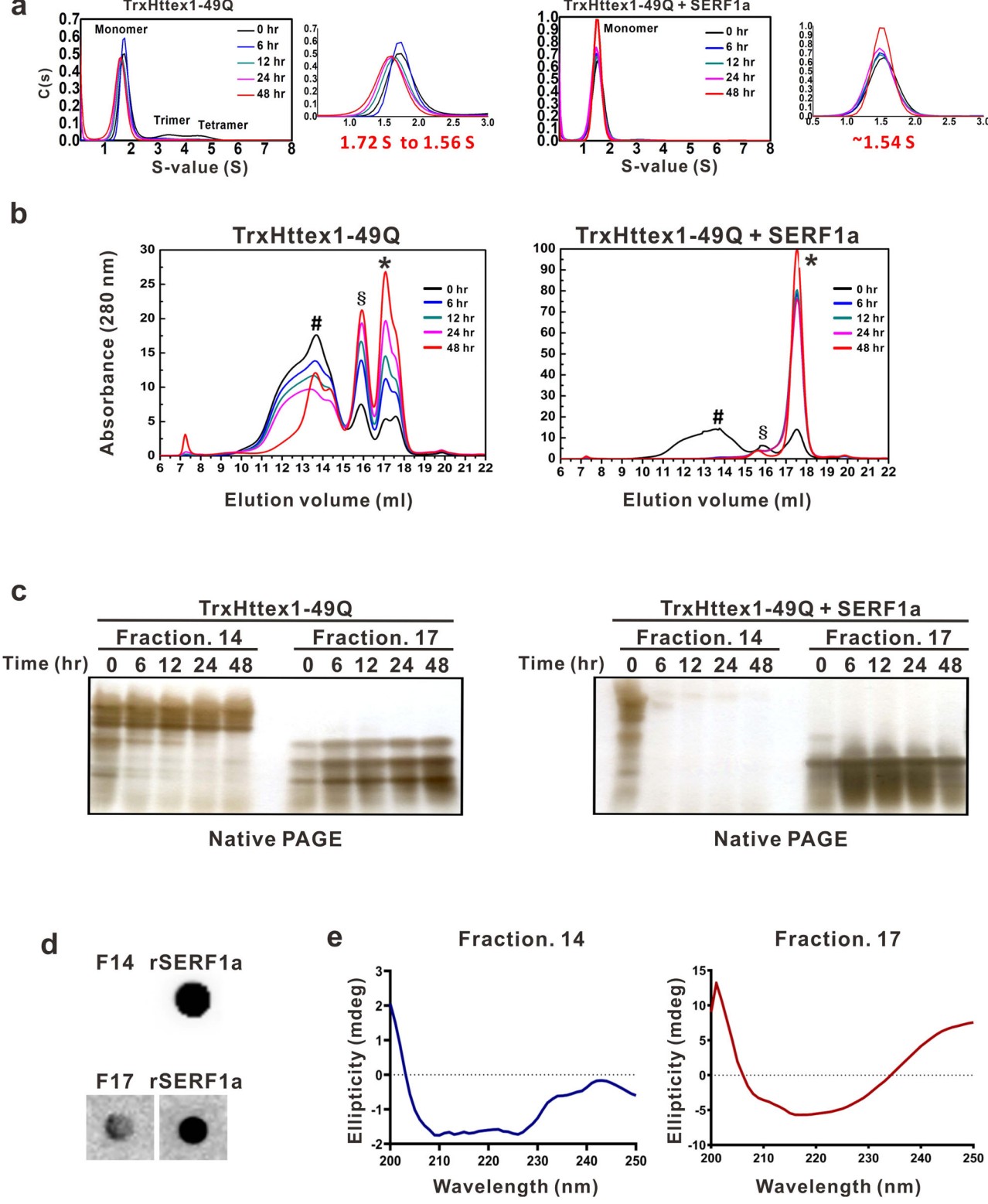

fluorescence microscopy (Fig. 6a). The cells were collected and lysed in Triton-containing buffer for filter trap assay (Fig. 6b). We found that overexpression of SERF1a increased 109Q aggregation as evidenced by fluorescence images and filter trap assay. Next, the cells were subjected to MTT assay to measure the cytotoxicity (Fig. 6c, Supplementary Data 1). The MTT reduction result showed that co-expression of EGFP-Httex1-109Q and SERF1a

greatly increased the cytotoxicity to ~29.7% compared with EGFP-Httex1-109Q alone at ~23.5%. The EGFP-Httex1-25Q with and without SERF1a caused ~19.2% and ~8.5% cytotoxicity, respectively, but without significant difference. Httex1-109Q showed higher toxicity than Httex1-25Q. Taken together, the results demonstrated that SERF1a plays a detrimental role to promote Httex1 cytotoxicity.

**Fig. 5 SERF1a induces conformational transition of TrxHttex1-49Q to a more homogeneous species.** TrxHttex1-49Q was incubated with and without equimolar SERF1a, and time-course samples were collected for experiments. **a** Analytical ultracentrifugation analysis of TrxHttex1-49Q at different incubation times. TrxHttex1-49Q with and without SERF1a was incubated, and samples at 0, 6, 12, 24, and 48 h were collected and subjected to sedimentation velocity (SV) experiments. Data were analyzed in a continuous c(s) distribution model, and sedimentation coefficients were obtained. **b** Size-exclusion chromatography (SEC) of TrxHttex1-49Q in the absence or presence of SERF1a. TrxHttex1-49Q was incubated with and without equimolar SERF1a, and time-course samples were collected at different timepoints for SEC. Peaks were labeled with different symbols #, §, and *. **c** Native PAGE for SEC fractions of TrxHttex1-49Q with and without SERF1a. Fractions 14 and 17 from SEC at different timepoints were collected for native PAGE and detected by silver staining. **d** Dot blot for SEC fraction 14 and 17 probed with SERF#1 (1:100). rSERF1a: recombinant SERF1a as a positive control. **e** Far-UV CD analysis for SEC fraction 14 and 17. Buffer background was subtracted from the spectra of fraction 14, and SERF1a alone spectra were subtracted from that of fraction 17.

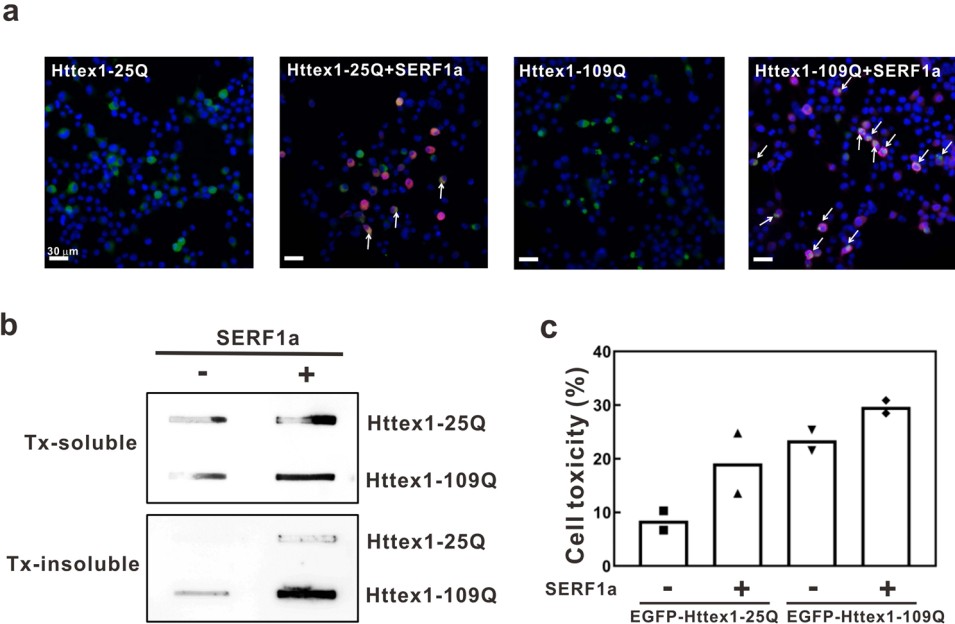

**Fig. 6 SERF1a promotes the aggregation and toxicity of Htt exon 1 in neuronal cell line. a** Neuro-2A cells transiently co-transfected with SERF1a-myc or myc-only control with EGFP-tagged huntingtin exon 1 (EGFP-Httex1-25Q or EGFP-Httex1- 109Q) for 48 h. Immunocytochemistry was performed with anti-myc antibody to detect SERF1a signal (red) and EGFP signal (green). Merged images were shown. Httex1 aggregates were indicated by arrows. The scale bars are 30 μm. **b** Cell lysates were collected and separated into Triton-soluble and -insoluble fractions and subjected to filter retardation assay. Httex1 aggregates retained on the filter membrane were detected by MW7 antibody against Htt. **c** Cytotoxicity of Httex1-25Q and Httex1-109Q-transfected N2a cells co-transfected with SERF1a or control. MTT reduction assay was performed in biological duplicate.

**SERF1a promotes Htt aggregation in iPSC-derived neurons from patients with HD.** Next, we employed iPSCs generated from fibroblasts of normal control (C1) and patient with HD (GM) and differentiated them to GABAergic neurons to confirm the SERF1a effect in neurons of patients with HD. The iPSC technology is a research frontier to provide authentic human cells for validation of the disease phenomenon. After differentiation for 12 weeks, the neurons were infected by lentivirus containing SERF1a fused with N-terminal EGFP. After 3 days of infection, we immunostained Htt aggregates recognized by MW7 antibody[34], specific to the polyproline region of Htt, in iPSC-derived neurons. The results showed that overexpression of EGFP-SERF1a significantly increased the Htt aggregation in the diseased neurons carrying mutant Htt (Fig. 7b). The percentage of Htt aggregates in SERF1a-positive neurons of GM strain was 17.54%, while that in control neurons was 4.84% (Fig. 7c, Supplementary Data 1). The neurons infected with EGFP controls (Fig. 7a) showed only low Htt signals in control and diseased neurons, i.e., 0.13% and 1.40%, respectively. The images of representative cells are shown in Supplementary Fig. 15. EGFP-SERF1a was expressed in the nucleus and cytoplasm, while Htt aggregates were only observed in the diseased iPSC-derived neurons. Expectedly, EGFP-SREF1a and Htt were highly colocalized. Therefore, human SERF1a was confirmed to promote mutant Htt aggregation in HD iPSC-derived neurons.

**SERF1a transcript and expression levels are elevated in HD subjects.** To further understand whether SERF1a in HD is different from the normal subjects, we performed real-time quantitative PCR (Q-PCR) on the brain lysate of HD transgenic mice (R6/2 mice) and human HD iPSCs, as well as enzyme-linked immunosorbent assay (ELISA) on plasma of HD patients (Fig. 8). From Q-PCR results, we found that both of HD transgenic mice (Fig. 8a, Supplementary Data 1) and human HD iPSCs (Fig. 8b, Supplementary Data 1) had higher *SERF1a* transcript level than the normal control, and especially that in human HD iPSCs was approximately two-fold increase compared to the control iPSCs. Moreover, to facilitate SERF1a detection, we generated a monoclonal antibody named SERF#1 by immunizing mice with a SERF1a C-terminus peptide and validated the antibody by ELISA and western blot with recombinant SERF1a (Supplementary Fig. 16). Then, we applied SERF#1 antibody to detect plasma of 18 samples from the HD patients and 18 samples from normal controls coated on the ELISA plate. The recombinant SERF1a spiked-in PBS and normal plasma were performed and the one

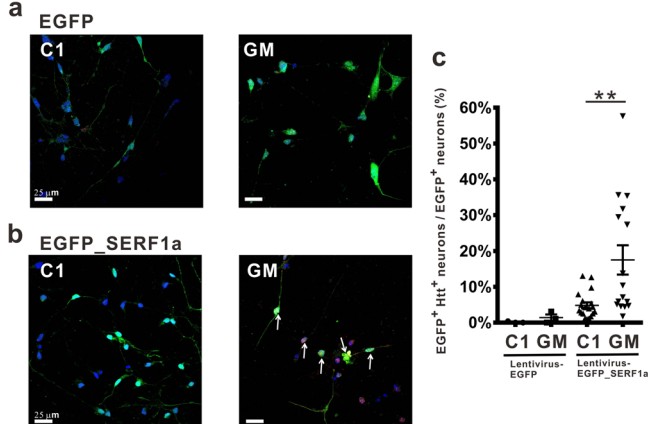

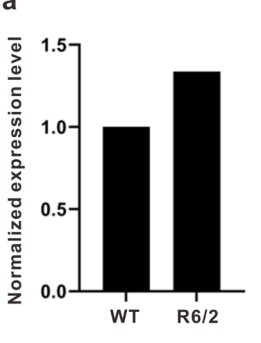

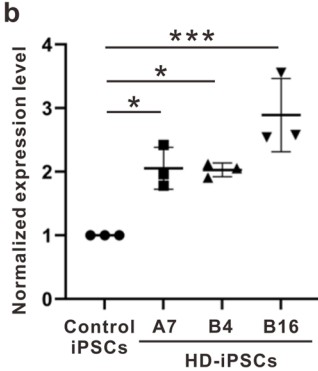

**Fig. 7 SERF1a enhances Htt aggregation in HD iPSC-derived neurons.**
Confocal microscopic images of iPSCs-derived neurons of normal control and HD patient infected with lentiviral EGFP construct (**a**) or EGFP_SERF1a construct (**b**). Cells were immune-stained with Htt antibody MW7. The colors for SERF1a (green), DAPI (blue), and Htt (red) are shown. The scale bars are 25 μm. (**c**) Calculated percentage of EGFP-positive (EGFP[+]) neurons harboring Htt aggregates (Htt+) in iPSC-derived neurons of normal control and patients with HD. The number of double-positive neurons was normalized to the total number of EGFP[+] neurons. Only cells with Tuj1 markers were counted. EGFP C1, $n = 3$; EGFP GM, $n = 3$; EGFP_SERF1a C1, $n = 19$; EGFP_SERF1a GM, $n = 16$. Data were presented as mean ± sem and analyzed by unpaired Student's t-test. **$P < 0.01$. The error bars represent SEM.

spiked-in normal plasma was used as a standard curve (Supplementary Fig. 17). The result showed that the HD patients had significantly higher SERF1a level, ~221.9 ng ml$^{-1}$, in the plasma compared to that in the normal control at ~ 152.4 ng ml$^{-1}$ (Fig. 8c, Supplementary Data 1). Together, these results strongly support a disease role of SERF1a in HD.

## Discussion

This study showed that SERF1a expedites and enhances TrxHttex1-polyQ fibril formation in a repeat length-dependent manner. SERF1a interacts with mutant TrxHttex1 majorly through α-helical regions, and the interaction is intensified by enhancing α-helical content. SERF1a predominantly interacts with N-terminus of Htt peptides. SERF1a facilitates converging the mutant TrxHttex1-polyQ monomeric conformers to a unique conformation. On the basis of previous literature, which demonstrated two monomeric Httex1-polyQ conformers[13], and the native PAGE result in the present study, in the presence of SERF1a, the monomeric mutant TrxHttex1-polyQ may be rapidly converted from α-helical structure to a β-sheet-rich species that accelerates the fibril formation. The results further validated that SERF1a promotes Httex1-polyQ aggregation and cytotoxicity in human neuroblastoma and induces significantly higher aggregation in HD iPSC-derived neurons than in healthy control. We also found that the transcript and expression level of SERF1a were increased in HD subjects, including HD transgenic mice, human HD iPSCs, and HD patients' plasma. On the basis of these findings and those of previous literature, we proposed a mechanism for SERF1a effect on mutant Httex1-polyQ. SERF1a mainly interacts with the helical NT17 domain of Httex1-polyQ proteins via its helical regions with a stoichiometry of 0.5; one SERF interacts with two Httex1 proteins. Such interaction facilitates Httex1-polyQ track to undergo conformational conversion to a fibrillization-prone β-sheet monomer. The β-sheet enriched monomer rapidly aggregates into amyloid fibrils.

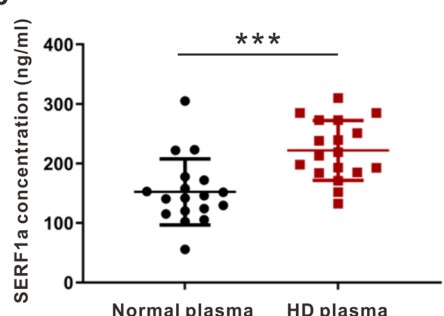

**Fig. 8 SERF1a is elevated in HD transgenic mice brain, HD iPSCs, and HD plasma. a** The *SERF1a* transcript level in whole brain of HD transgenic mice and wild-type littermate was analyzed by Q-PCR. **b** The *SERF1a* transcript level in human HD and control iPSC cells was analyzed by Q-PCR. Data were analyzed by one-way ANOVA, *$P < 0.05$, ***$P < 0.001$; $n = 3$. The error bars represent standard deviation. **c** SERF1a level is significantly higher in plasma of HD comparing to that in the normal control. Spike-in plasma was used as a standard to calculate SERF1a concentration. Normal plasma, $n = 18$; HD plasma, $n = 18$. Data were analyzed by unpaired Student's t-test, ***$P < 0.001$. The error bars represent standard deviation.

Previous reports revealed that the N-terminal region of Htt triggers rapid polyQ aggregation[43,44]. PolyQ fusion to NT17 induces NT17 to a more extended conformation in a repeat length-dependent manner that greatly enhances its aggregation[43]. NT17 domain is spatially close to the proline-rich region with polyQ lengths less than 32 repeats, but not with that more than 37 repeats[45]. Therefore, SERF1a binds to the NT17 of mutant Httex1 but not the NT17 of normal Httex1, may be due to conformational changes in NT17. This molecular mechanism is different from the Hsc70-NT17 binding mechanism, where Hsc70 binding to NT17 suppressed polyQ aggregation without polyQ length dependence[46]. In our case, SERF1a associated with NT17 with a ratio of 1:2 allows the development of NT17 dimers to enhance aggregation, which is different from the proposed single binding of Hsc70 and NT17. Besides, the effect of poly-Q length may only occur upon binding to smaller NT17 binding proteins such as SERF1a, ~7 kDa, than larger binding proteins like Hsc70, ~71 kDa.

Coiled-coil structure plays a critical role in polyQ protein aggregation, and many Htt binding partners have coiled coils regions[23]. The possible coiled-coil regions in SERF1a and Httex1 were characterized, and the propensity was analyzed using the program DrawCoil[41]. The prediction showed that the N-terminus of Httex1 (residues 4–40) and the C-terminus of SERF1a (residues 39–62) have high CC probability (Supplementary Fig. 18). Here, SERF1a interacted with Httex1-39Q through α-helical regions by NMR study, and the interaction was enhanced with

increased helical contents by TFE treatment using intrinsic fluorescence titration, indicating a possible coiled-coil interaction between NT17 and SERF1a. Inversely, the coiled-coil interaction was disrupted by destabilizing α-helix structure of NT17 region, as shown in the Htt peptide study. Together, our results suggest that SERF1a interacts with mutant Httex1 by coiled-coil interaction.

In the Htt peptide study, SERF1a was shown to predominately interact with NT17 and Htt-3 but not the ones with proline mutations in NT17 region (Htt-2, Htt-4, and Htt-5). NT17 showed over fourfold stronger affinity to SERF1a than Htt-3, showing that polyQ region influences the interaction. However, SERF1a did not bind to Htt-0, possibly due to the dimerization of Htt-0 and/or the short polyQ in Htt peptides. Similarly, Htt-1 formed pentamers and octamers that may hindered the binding to SERF1a. In fact, NT17 was shown to adopt different conformations when attached to short and long polyQ tract[43]. Therefore, SERF1a and Httex1 interaction may be facilitated due to conformational changes of NT17 in long poly tract. However, whether full-length Httex1 with extended polyQ interacts with SERF1a through NT17 needs to be further investigated.

Htt exon1 has been shown to fold into distinct misfolded monomers, leading to different aggregates[10,47]. A previous report has revealed that the expanded polyQ monomers adopt two conformations, including α-helix-dominant and β-sheet-rich structures[13], in which the β-sheet conformer migrates faster than the α-helix form in native PAGE. They proposed that native polyQ monomers undergo a conformational change from α-helix form to β-sheet form that is more prone to aggregation and more toxic. In addition, a compact β-sheet structure of mutant Htt exon1 was found to induce toxicity in mammalian neuronal cells and primary neuronal cultures[13,48,49]. The present results demonstrated that monomeric mutant TrxHttex1, as evidenced by AUC result, eluted into three peaks in SEC, indicating possibly different conformers of Httex1 monomers; in the presence of SERF1a, the peaks converged to a single species. Combined with far-UV CD spectra of the isolated fractions and the native PAGE results similar to previous literature[13], the β-sheet conformer that migrated faster in native PAGE was enriched in the presence of SERF1a. Therefore, we proposed that SERF1a enhances the Httex1 conformational conversion to form an amyloidogenic monomeric species enriched in β-sheet. However, it should be noted that Trx-fusion tag was not removed from Httex1 protein for most experiments in this study. Since Trx can retard protein aggregation and was fused to the N-terminus of Httex1 where SERF1a binds to, it may have some effects on the interaction between SERF1a and Httex1. The results investigated here may not totally represent the actual Httex1 protein.

PolyQ-associated diseases, including HD, are devastating and fatal neurodegenerative diseases without effective treatment. SERF was found to promote amyloid formation in vitro and in vivo[25,27]. It was shown that SERF1a directly interacts with α-synuclein to facilitate the generation of on-pathway aggregates, and then enhances α-synuclein fibrillization. Interestingly, SERF1a was also identified as an RNA-organizing protein forming fuzzy complexes with RNA that leads to its inability of distinguishing nucleic acid and amyloid protein[28]. These studies increased the role of SERF1a in amyloid formation and neurodegenerative diseases. Unlike molecular chaperones that rescue or prevent protein misfolding, SERF promotes several protein aggregations to amyloid formation, representing a new class of modifiers for the field of protein misfolding to be intensively investigated. By understanding the modifier mechanism towards protein aggregation, new knowledge could be gained and novel therapeutic strategies against neurodegenerative diseases could be developed.

## Methods

**Recombinant SERF1a preparation**. Human SERF1a gene (GeneBank accession number BC021174) was purchased from Bioresource Collection and Research Center (BCRC number g1004029D09). Full-length SERF1a cDNA was amplified by PCR. SERF1a cDNA was double digested with NcoI and BamHI, ligated into pET14b (Novagen), and expressed in *E. coli* BL21(DE3). The construct encodes His-SERF1a with a thrombin cutting site between His-tag and SERF1a. Bacteria were grown in 1 L of LB broth containing ampicillin at 37 °C, with shaking and inducing by IPTG. Cells were harvested by centrifugation. Cell extract was disrupted on suspension buffer containing 20 mM Tris-HCl (pH 7.9), 5 mM imidazole, 0.5 M NaCl, and cocktailed protease inhibitor. Supernatant was loaded onto a Ni-Sepharose column (HisPrep FF-Ni Sepharose 6 Fast Flow; GE Healthcare Biosciences). After washing with 10 column volume of the suspension buffer, the column was developed with a linear imidazole gradient from 5 mM to 500 mM in the same buffer. Fractions containing His-SERF1a were pooled and concentrated by ultrafiltration and dialyzed against thrombin cutting buffer (20 mM Tris-HCl, pH 8). The dialyzed product was concentrated and cleaved with thrombin. SERF1a was further purified by FPLC using the Mono-S column with a linear gradient of 5 mM to 500 mM NaCl in buffer. Fractions containing SERF1a were pooled, dialyzed against 50 mM Tris-HCl (pH 6.8) and 20 mM NaCl, and stored at 4 °C.

**Recombinant TrxHttex1-polyQ preparation**. Different glutamine (Q) lengths of Htt-polyQ exon 1 were built up from pcDNA3.1-Htt-(Q)25-hrGFP plasmid by Site-directed Mutagenesis (KAPA Biosystems). PCR products were subcloned to pCR2.1-TOPO-TA vector (Invitrogen technology) for amplification and confirmed by DNA sequence. Httex1-polyQ cDNA was double digested with NcoI and EcoRI, ligated into pET32a (Novagen), and expressed in *E. coli* Rosetta 2 (DE3). The construct encodes TrxHttex1-polyQ with a thrombin cutting site between thioredoxin-tag and Httex1-polyQ. Bacteria were grown in 1 L of LB broth containing ampicillin at 37 °C, with shaking and inducing overnight at 25 °C by IPTG. Cells were harvested by centrifugation. Cell extract was disrupted on suspension buffer containing 20 mM Tris-HCl (pH 7.9), 5 mM imidazole, 0.5 M NaCl, and protease inhibitor. Supernatant was loaded onto a Ni-Sepharose column (HisPrep FF-Ni Sepharose 6 Fast Flow). After washing with 10 column volume of the suspension buffer, the column was developed with a linear imidazole gradient from 5 mM to 500 mM in the same buffer. Fractions containing TrxHttex1-polyQ were pooled and concentrated by ultrafiltration, and dialyzed to Mono-Q buffer (20 mM Tris-HCl, pH 8, 0.2 mM EDTA, 0.5 mM DTT, 5 mM NaCl, 0.02% NaN₃). After dialyzing, protein was further purified by FPLC using the Mono-Q column with a linear gradient of 5 to 500 mM NaCl in buffer. Fractions containing TrxHttex1-polyQ were pooled, dialyzed against 10 mM Tris-HCl (pH 8.0), and stored at −80 °C. Httex1-polyQ proteins were produced as above, except for cleaving with thrombin. Httex1-polyQ proteins were purified by FPLC using HisTrap Sepharose column to remove thioredoxin tag.

**Htt peptide preparation**. NT17, Htt-0, Htt-1, Htt-2, Htt-3, and Htt-4 peptides were synthesized by the peptide synthesis core in Genomics Research Center Academia Sinica. Htt-5 peptide was synthesized by Scientific Biotech. The peptides, 0.1–0.2 mg for ITC and AUC and 0.3–1 mg for SAXS, were treated with 20 μl of 100% trifluoroacetic acid (TFA) at room temperature overnight to dissolve preformed aggregates. TFA was then removed by SpeedVac vacuum concentrator. The dried peptides were dissolved by 10 μl of 1% TFA and added into 480 μl of 10 mM PB (pH 7.4) with 16.5 μl of 100 mM NaOH. All peptides were centrifuged to exclude precipitates, and the concentration of stocks was quantified by bicinchoninic acid (BCA) assay.

**Thioflavin T (ThT) Assay**. Different expanded TrxHttex1-polyQ proteins were centrifuged at $17,000 \times g$ at 4 °C for 15 min to remove precipitates, and supernatant was quantified. Different expanded TrxHttex1-polyQ proteins (50 μM) were mixed with and without SERF1a at equimolar ratio and 10 μM ThT. Samples were constantly rotated at 200 rpm and incubated in a 96-well ELISA plate at 37 °C. For SERF1a and α-synuclein aggregation assay, α-Synuclein at 50 μM in Tris buffer (10 mM Tris-HCl, pH 7.4, and 150 mM NaCl) was incubated with different SERF1a concentrations in a 384-well ELISA plate. ThT at 10 μM was added in the samples. The samples were constantly rotated at 400 rpm during the incubation. ThT fluorescence was measured at 485 nm with an excitation of 442 nm at 25 °C by a microplate reader (SpectraMax M5; Molecule Devices) with SoftMax Pro 5.4. The data from independent trials were averaged, and the standard deviations were calculated.

**Filter trap assay**. A 100 μl aliquot of end-point products of aggregated samples was loaded onto a Bio-Dot SF microfiltration apparatus (Bio-Rad, Hercules, CA, USA) equipped with an in-house vacuum system. Cellulose acetate membranes with 0.2 μm (Advantec) were blocked with 5% skim milk in TBS buffer and incubated with MW7 (1: 3,000; DSHB) antibody at 4 °C overnight. The secondary antibody was horseradish peroxidase conjugated anti-mouse IgG (1:5,000; Millipore, Billerica, MA, USA), and it probed for 2 h at room temperature. The membrane was developed with electrochemiluminescence reagent (Millipore) and the signals were detected by ImageQuant LAS 4000. For cell lysate, Neuro-2A cells

(ATCC, Cat#CCL-131) were seeded at $2.5 \times 10^5$ cells per well at six-well plate for 24 h and transiently co-transfected with SERF1a-myc or myc-only control with EGFP-tagged huntingtin exon 1 (EGFP-Httex1-25Q or EGFP-Httex1-109Q) for 48 h. Pellets were collected and suspended in ice-cold buffer (10 mM PBS, pH 7.5, 1% Triton X-100, and cocktailed protease inhibitor) and homogenized. After centrifugation at 1,000 g for 5 min at 4 °C, the supernatant was collected into new tubes, and the pellet was resuspended in 100 μl of resuspending buffer (10 mM Tris-HCl, pH 8.0, 4 mM EDTA, 4% SDS, 100 mM DTT, 500 μM CaCl$_2$, and 5 mM MgCl$_2$) and boiled for 5 min. Triton-soluble supernatant and triton-insoluble fraction were applied to cellulose acetate membranes with 0.2 μm (Advantec) by loading onto a Bio-Dot SF microfiltration apparatus (Bio-Rad, Hercules, CA, USA) equipped with an in-house vacuum system. The membranes were then blocked with 5% skim milk in TBS buffer and incubated with MW7 (1:3,000; DSHB) antibody at 4 °C overnight. The secondary antibody was horseradish peroxidase-conjugated antimouse IgG (1:5,000; Millipore, Billerica, MA, USA), and it was probed for 2 h at room temperature. The membranes were developed by electro-chemiluminescence reagent (Millipore) and the signals were detected by Image-Quant LAS 4000.

**Transmission electron microscopy (TEM)**. The end-point products of aggregated samples were placed on glow-discharged, 400-mesh Formvar carbon-coated copper grids (EMS Inc., Hatfield, PA, USA) for 3 min (Aβ or α-synuclein) or for 1 min (TrxHttex1-polyQ), rinsed, and negatively stained with 2% uranyl acetate (UA). Samples were examined in Hitachi H-7000 TEM (Hitachi Inc., Tokyo, Japan) at an accelerating voltage of 75 kV. For immunogold staining, the incubated TrxHttex1-39Q (50 μM) with and without equimolar SERF1a samples were dropped on the grids for 10 min and rinsed. After air-drying, the grids were then probed with primary SERF1a antibody (1:100; monoclonal antibody) and the 10 nm gold-conjugated secondary anti-mouse IgG antibody (1:500; abcam). The grids were finally stained with 1% UA. The samples were observed with a FEI Tecnai G2 F20 S-TWIN transmission electron microscope at 120 kV with TEM user interface and DigitalMicrograph software.

**Dot blotting**. Fifty μM of each expanded TrxHttex1-polyQ with and without equimolar SERF1a were incubated at 37 °C with continuous shaking at 200 rpm. Samples (4 μl) were collected at different timepoints and dotted into nitrocellulose membrane with two replicates. Membranes were blocked with 5% skim milk in TBS buffer and incubated with 1C2 (1:3,000; Millipore), MW7 (1:3,000; DSHB), A11 (1:1,000; Invitrogen), or OC (1:10,000; Millipore) antibodies separately at 4 °C overnight. The secondary antibody was horseradish peroxidase conjugated anti-mouse or anti-rabbit IgG (1:5,000; Millipore, Billerica, MA, USA), and it was probed for 2 h at room temperature. The membranes were developed with electro-chemiluminescence reagent (Millipore) and the signals were detected by Ima-geQuant LAS 4000.

**Far-UV CD spectroscopy**. For time-course experiment, fifty μM of each expanded TrxHttex1-polyQ with and without equimolar SERF1a were incubated at 37 °C with continuous shaking at 200 rpm and measured at different timepoints. For TFE treatment, fifty μM of SERF1a, TrxHttex1-polyQ, and Htt peptides in 10 mM PB buffer were measured in the presence of various TFE concentrations. The α-helical content was calculated by signal at 222 nm divided by signal at 208 nm and plotted against TFE concentration.Far-UV spectra were recorded from 250 nm to 195 nm with a J-815 CD spectropolarimeter (Jasco Inc., Easton, MD, USA) with Spec-traManager 2. Measurements were performed at room temperature.

**Intrinsic fluorescence spectroscopy**. Intrinsic fluorescence was excited at 257 nm, and fluorescence emission was recorded at 282 nm. TrxHttex1-polyQ or Httex1-polyQ at 25 μM was titrated with 800 μM SERF1a to final concentrations in the range of 0–100 μM. All experiments were conducted at 25 °C in a circulating water bath by using FluoroMax-3 spectrofluorometer (Horiba Jobin Yvon, Kyoto, Japan) with FluorEssence V3.5.

**NMR Spectroscopy**. $^{15}$N-labeled proteins were expressed in M9 minimal media containing $^{15}$NH$_4$Cl and glucose. Purified $^{15}$N-labeled SERF1a was concentrated to 0.1 mM in 50 mM Tris-HCl (pH 6.8), 20 mM NaCl, and 3 mM NaN$_3$ for NMR structural studies. $^{15}$N-SERF1a alone as reference was 70 μM and titrated with TrxHttex1-39Q at the indicated ratio. All NMR experiments were carried out at 298 K on Bruker Avance 850 MHz NMR or 600 MHz spectrometers equipped with 5 mm triple resonance cryoprobe and Z-gradient. Data were acquired and processed using the software Topspin2.1 (Bruker, Germany) and further analyzed with Sparky version 3.114 (Goddard and Kneller). 1H chemical shifts were externally referenced to 0 ppm of 2,2-dimethyl-2-silapentane-5-sulfonate, and $^{13}$C and $^{15}$N chemical shifts were indirectly referenced in accordance with IUPAC recommendations[50]. Protein backbone resonance assignments were based on standard triple resonance experiments[51]: HNCACB, CBCA(CO)NH, HNCO, HN(CA)CO, HNCA, and HN(CO)CA. The chemical shift perturbation for combined $^1$H and $^{15}$N resonances of SERF1a was calculated using the following equation: $\Delta ppm = [(5 * \Delta^1H)^2 + (\Delta^{15}N)^2]^{1/2}$ [52]. The intensity drop rate was normalized to K62, which shown the least intensity drop, that is, I (bound)/ I (free) of K62 was assumed to be 1.00.

**Isothermal titration calorimetry (ITC)**. MicroCal iTC200 (GE) was used for ITC experiments. SERF1a was in 10 mM PB, pH 7.4. For Htt-0, Htt-1, Htt-2, Htt-4, and Htt-5, 300 μM SERF1a in syringe was injected into the cell containing 30 μM Htt peptides. The volume of each injection was 2 μl. For Htt-3 and NT17, 250 μM SERF1a in syringe was injected into the cell containing 50 μM Htt peptides. The volume of each injection was 1.5 μl. The cell of the calorimeter was maintained at 26 °C. ITC analysis software MicroCal Analysis Launcher (GE Healthcare) was used for data analysis.

**Small-angle X-ray scattering (SAXS)**. Htt peptides, 0.3–1 mg, were treated with 20 μl of 100% TFA at room temperature overnight to dissolve preformed aggregates. TFA was then removed by SpeedVac vacuum concentrator. The dried peptides were dissolved by 10 μl of 1% TFA and added into 480 μl of 10 mM PB (pH 7.4) with 16.5 μl of 100 mM NaOH. SERF1a was in 10 mM PB, pH 7.4. NT17 at 0.255 mg ml$^{-1}$ and 129.2 μM, SERF1a at 0.445 mg ml$^{-1}$ and 60.7 μM, and the mixture in 2:1 molar ratio were prepared for SAXS measurements. Htt-3 at 0.8 mg ml$^{-1}$ and 203.3 μM, SERF1a at 0.7 mg ml$^{-1}$ and 95.4 μM, and the mixture in 2:1 molar ratio were prepared for SAXS measurements. SAXS data were measured at TPS 13 A BioSWAXS end station by using a 15 keV beam and an Eiger X 9 M detector[53,54]. The sample solutions loaded onto a size exclusion column (SEC) with a flow rate of 0.35 ml/min at 10 °C for SEC-SAXS. The eluate from the SEC was directed to the quartz capillary (2 mm dia. and a wall thickness of 20 μm) of the SEC-SAXS system for X-ray exposure continuously with 2 s per frame over the elution peak. The frame data of well-overlapped SAXS profiles were averaged and subtracted with buffer scattering using the TPS 13 A SWAXS Data Reduction Kit (Ver. 3.6), and analyzed using ATSAS. The radii of gyration $R_g$ were extracted, and DAMMIN model fittings were performed using the ATSAS package[55].

**Analytical ultracentrifugation (AUC)**. Sedimentation velocity (SV) experiment in AUC was conducted on a Beckman Optima XL-I analytical ultracentrifugation (Beckman Coulter, USA). By using an An-60Ti rotor, SERF1a at 50 μM in 10 mM PBS (pH 7.0) was centrifuged at 60,000 rpm for 72 h at 4 °C or 20 °C. The moving boundary under the absorption at 229 nm was measured with a scanning recorder every 4 min. TrxHttex1-polyQ at 50 μM in 20 mM Tris-HCl (pH 8.0) were centrifuged at 40,000 rpm for 24 h at 20 °C. The moving boundary under the absorption at 280 nm was measured with a scanning recorder every 2 min. For Htt peptides, all peptides in 10 mM PB, pH 7.4 were centrifuged at 42,000 rpm for 24 hr at 25 °C by using an An-60Ti rotor. The moving boundary was monitored continuously under the absorbance at 220 nm for NT17 and Htt-0, 225 nm for Htt-1 and Htt-2, and 230 nm for Htt-3, Htt-4, and Htt-5. The parameters for data analysis were determined by using SEDNTERP (NIH) and SV results were analyzed by SEDFIT (U.S. NIH) with C(S) distribution method.

**Size-exclusion chromatography**. Samples at 50 μM with different timepoints were collected and centrifuged at 17,000 × g for 5 min, and 100 μl supernatants were injected into Superdex 200 10/300 gel-filtration column (GE Healthcare BioSciences). The column was calibrated with eight standard proteins of known molecular mass. The fraction samples were eluted with 10 mM Tris-HCl (pH 8.0) and 150 mM NaCl buffer at a flow rate of 0.4 ml min$^{-1}$. Absorbance at 280 nm was monitored.

**Native PAGE and silver staining**. SEC fractions were subjected to 4%–16% acrylamide gel for native PAGE and then stained by silver stain with the use of SilverXpress Silver Staining Kit (Invitrogen).

**Immunocytochemistry**. Neuro-2A cells (ATCC, Cat#CCL-131) were seeded at $2.5 \times 10^5$ cells per well at six-well plate for 24 h and transiently co-transfected with SERF1a-myc or myc-only control with EGFP-tagged huntingtin exon 1 (EGFP-Httex1-25Q or EGFP-Httex1-109Q) for 48 h. Cells were collected and subjected to fluorescence imaging. Fluorescent immunocytochemistry was performed on the prepared slides with primary antibody against c-myc (1:2,000; mouse, M4439-100UL, Sigma) and Alexa594 labeled anti-mouse (1:1,000; A-21125, Invitrogen) as the secondary antibody. Slides were imaged by Leica Automatic Upright Microscopy PM 6000B with HC PL APO 20×/0.7 objective lens by using a charge-coupled device camera Zyla 4.2 (Andor Technology Ltd) attached. For iPSC-derived neurons, the control- and HD-iPSC-derived neurons were prepared after 12 weeks of differentiation and infected with lentivirus containing EGFP control and EGFP-SERF1a for 3 days. After infection, neurons were maintained in B27/N2 medium. The cells were fixed in 4% paraformaldehyde and permeated by 0.5% Triton X-100 in PBS for 15 min at room temperature. Immunocytochemical staining was performed with anti-Htt antibodies MW7 (mouse; obtained from the Developmental Studies Hybridoma Bank; 1:1,000). Images of normal control and HD cells were acquired using confocal settings by confocal microscopy (Leica TCS-SP5-MP-SMD). Samples were observed under 10× HC PL Fluotar air objectives and 63× HC PL Apo oil CS2 objectives equipped with Leica TCS-SP8-MP-SMD confocal microscope with 405 nm diode laser for DAPI, argon 488 nm laser for EGFP-SERF, helium-neon 594 nm laser, and DPSS laser excited at 561 nm for Htt. Images were processed using LAS AF Lite 2.4.1 software (Leica Microsystems, Singapore). The number of Htt aggregates colocalized with overexpressing SERF1a in

cell was analyzed by Image J[56]. The number of Htt and SERF-positive neurons were normalized to total SERF positive neurons. Only Tuj1 positive cells were counted. Data were analyzed by unpaired Student t-test on SPSS (IBM, Armonk, New York, USA). **$P < 0.01$.

**Cytotoxicity assay**. Neuro-2A cells (ATCC, Cat#CCL-131) were seeded at $2 \times 10^4$ cells per well in 96-well plate for 24 h and transiently co-transfected with SERF1a-myc or myc-only vector control and EGFP-tagged huntingtin 25Q or 109Q at 37 °C for 48 h. MTT solution (Sigma) was added to each well and incubated for another 4 h. The medium was removed, and 100 μl of DMSO was added to dissolve the formazan crystals. Absorbance was measured at 570 nm, and the background signals caused by samples without cells were subtracted. Data were normalized with buffer control. MTT reduction was calculated to indicate cytotoxicity. The data was analyzed by GraphPad Prism9.

**Cultivation and neural differentiation of iPSCs**. Human control iPSC were derived from normal subjects (CON1; C1)[57]. HD-iPSC (GM23225; GM) derived from HD patient with 72 CAG repeats and purchased from Coriell Institute (Camden, NJ). iPSCs were maintained, subcultivated and differentiated into GABAergic neurons according to our published protocol[57]. In brief, iPSCs were induced into neural progenitor cells (NPCs) in N2 medium (DMEM/F12, 1X N2 supplement, 1% NEAA, 1 mM sodium pyruvate, 2 mM L-glutamine) supplemented with 20 ng/ml bFGF, 10 μM SB431542 and 100 nM LDN193189. The NPCs were subcultivated and maintained in N2 medium supplemented with 20 ng/ml bFGF. For GABAergic neuron differentiation, NPCs were cultured in N2B27 medium (DMEM/F12: Neurobasal medium (1 : 1), 0.5X N2, 0.5X B27, 1% NEAA, 0.5 mM sodium pyruvate, 2 mM L-glutamine) supplemented with 10 ng/mL bFGF for 4 weeks and medium was changed to B27 medium (Neurobasal medium, 1X B27, 1% NEAA, 2 mM L-glutamine) for terminal differentiation for 4 to 6 weeks. The neurons at 12 weeks of differentiation were used for lentivirus infection.

**Q-PCR**. Real-time quantitative PCR was performed from total RNA extraction of the whole brain of HD transgenic mice (R6/2) and the wild type littermate and iPSC following manufacture's protocol (KAPA SYBR® Fast qPCR kit) by LightCycler®480 (Roche) with Software release 1.5.0 SP4. The whole brain of HD transgenic mice (R6/2) and the wild type littermate was provided by Dr. Yijuang Chern, Academia Sinica with Academia Sinica IACUC approval. The primers for mouse SERF1 were qPCR-F (TGGAAATCAAAGAGAAATTGCC) and qPCR-R (GCTACCTTCTGCTTTTGTTGC). iPSCs were provided by Dr. Hung-Chih Kuo, Academia Sinica. The primers for human SERF1A were qPCR-F (TGGAAATC AACGAGAACTTGC) and SERF1A qPCR-R (GCTGCCTTCTGCTTTTCTTG). The data were normalized with GAPDH.

**Plasma sample**. Human plasma samples from healthy individuals and HD patients were collected from Chang Gung Memorial Hospital at Linkou, Taiwan. All relevant ethical regulations were followed and informed consent was obtained.Samples were fully encoded to protect patient confidentiality. The sample collections were approved by IRB in Chang Gung Memorial Hospital at Linkou and Academia Sinica (IRB01-12137).

**Generation of SERF1a antibody**. We generated a monoclonal antibody named SERF#1 by immunizing mice with SERF1a C-terminus peptides (EKQKAA-NEKKSMQTREK) following monoclonal antibody production in mice which were injected every 2 weeks for 6 injections in total by LTK BioLaboratories, Taiwan. The monoclonal antibody was collected from the media of hybridoma and purified by Protein G Sepharose 4 Fast Flow beads. The affinity and specificity were validated by ELISA and Western blot with recombinant SERF1a.

**Enzyme-linked immunosorbent assay (ELISA)**. One hundred μl of 10-fold PBS-diluted normal and HD plasma were coated in a 96-well ELISA plate and incubated at 4 °C overnight. After removing the plasma, the plate was washed by TBST and blocked with 5% BSA in TBS at room temperature for 2 hr. The plate was then probed by the primary SERF#1 antibody (1:100) at 4 °C overnight followed by the secondary HRP-conjugated anti-mouse antibody (1:5,000; GTX213111-01, Gene-Tex) at room temperature for 1 hr. The signal was finally developed by TMB microwell peroxidase substrate and the reaction was stopped by adding 250 mM HCl. The absorbance at 450 nm was recorded by using SpectraMax M5 microplate reader (Molecular Devices) with SoftMax Pro 5.4. The data was analyzed by GraphPad Prism9.

**Recombinant α-synuclein preparation**. α-Synuclein was provided by Dr. Winny Ariesandi, GRC, Academia Sinica[58]. In brief, the protein was expressed in E. coli and extracted by periplasmic osmotic procedures without heating. Then, it was loaded onto Q Sepharose FF column with a salt gradient from 100 mM to 500 mM NaCl. Fractions containing α-synuclein were purified through spin filter with 30 kDa MWCO (Millipore, USA), and the protein was collected in the flow through. The protein was dialyzed to 10 mM Tris-HCl at pH 7.4 for experiments.

**Aβ preparation**. Aβ was synthesized by solid-phase peptide synthesis in the Genomics Research Center, Academia Sinica, Taiwan[59]. Lyophilized Aβ40 peptide (0.1 mg) was dissolved in hexafluoroisopropanol (HFIP) and evaporated HFIP in vacuum to prepare Aβ stock. Aβ peptide was dissolved by anhydrous dimethyl-sulfoxide (DMSO) and then diluted in 100 μl Tris buffer (10 mM Tris-HCl, pH 7.4, and 150 mM NaCl). The solution was quantified by absorbance at 280 nm ($\varepsilon = 1280 \text{ cm}^{-1}\text{M}^{-1}$) and used as a stock solution to prepare Aβ at 25 μM for all experiments.

**Western blot**. The incubated samples were centrifuged at 15,000 rpm for 10 min to separate the soluble proteins in supernatant and the insoluble fibrils in pellet. Each fraction was subjected to a 13% Tris/tricine separating gel with 10% spacing gel and 4% stacking gel for SDS-PAGE and transferred to PVDF membrane (GE) which was then probed with the primary antibody SERF#1 (1:100) and the secondary HRP-conjugated anti-mouse antibody (1:5,000; GTX213111-01, GeneTex). The membrane was then developed with ECL reagent (Millipore) and the signals were detected by ImageQuant LAS 4000.

**Statistics and reproducibility**. Statistical tests on all data were performed using GraphPad Prism 9. For the calculated percentage of Htt aggregates in iPSCs-derived neurons (Fig. 7c), the data is presented as the mean ± SEM and analyzed by unpaired Student's t-test. For Q-PCR (Fig. 8b), the data is shown as the mean ± SD and statistical significance was determined by one-way ANOVA. For ELISA (Fig. 8c), the data is shown as the mean ± SD and statistical significance was determined by unpaired Student's t-test. All bar graphs are representative of the experiments with n < 3. Statistical significance was applied for the experiments with n > 3.

**Reporting summary**. Further information on research design is available in the Nature Portfolio Reporting Summary linked to this article.

## Data availability
All data supporting the findings of this study are available within the paper and its Supplementary Information. The source data is provided in Supplementary Data 1. The uncropped gel and blots are available in Supplementary Figs. 19–22.

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

## Acknowledgements

We thank Dr. Winny Ariesandi, GRC, Academia Sinica, for providing α-synuclein and Dr. Yijuang Chern, IBMS, Academia Sinica for providing Httex1-25Q and Httex1-109Q plasmids and mouse brain tissues. We thank Mr. Yao-Kwan Huang in the TEM core facility and Ms. Yi-Ping Huang in NMR core facility, Academia Sinica, for their assistance on TEM and NMR, respectively. We thank Academia Sinica to fund the research.

## Author contributions

T.Y.T. performed the ITC and ELISA experiments, wrote the manuscript, and revised the manuscript. C.Y.C. performed most experiments and wrote the first version of the manuscript. T.W.L. and M.Y.C. performed ICC of iPSC-derived neurons. F.L.C. and H.C.K. provided iPSC-derived neurons and wrote the method for cultivation and neural differentiation of iPSCs. T.C.L., O.S., A.C.S., and U.S.J. performed SAXS experiments, analyzed the data, and wrote the SAXS part of the manuscript. Y.C.L. performed mouse qPCR experiment. C.M.C. provided human plasma. C.F.C. contributed to NMR data analysis and wrote the method for NMR. Y.R.C. conducted the research and edited the manuscript.

## Competing interests

The authors declare no competing interests.

**Additional information**

