## [Peer Review File · Communications Biology]

Reviewers' comments:

Reviewer #1 (Remarks to the Author):

The revised version of the manuscript "Amyloid modifier SERF1a interacts with PolyQ-expanded huntingtin-exon1 via helical interactions and exacerbates polyQ-induced toxicity" addresses the majority of points that I raised before. I appreciate the efforts to improve the study, with an important number of new experiments. Despite these efforts, I am not fully convinced about the highest resolution details of the mechanism based on the interpretation of the NMR and SAXS data.

- I am not convinced that the interpretation of the massive decrease of the SERF1a NMR intensities upon addition of HttEx1 is correct. It is true that the HttEx1 construct is large... but it is also true that it is highly flexible. Thus, I would not expect a systematic intensity decrease for all residues, including the most terminal ones. Maybe it would help if the intermediate titration point was shown, to see a dose dependent effect on the NMR intensities

- The SAXS data are also intriguing. I appreciate the efforts to measure new SEC-SAXS data, which should be described in the Methods section. The full SEC profile should be shown. It is unclear that for mixtures, the complex is captured taking into account the dilution caused by the column and the transient nature of the interaction. It could be that the authors capture an unbound mixture of both peptides, which would explain the small R_g found for the mixture when compared with SERF1a alone. I think that the authors should try to address these points and be less conclusive with respect to the NMR and SAXS experiments. The rest of the data seem solid and relevant.

Reviewer #2 (Remarks to the Author):

Response to reviewers 2 and 3.

The relevance mentioned by reviewer 2 was addressed.

Reviewer 3. Question 1.

The concerns regarding binding of SERF1a to beta sheet containing stages of Httex1 aggregation are not well addressed.

Figure 5 presents data showing that SERF1 preferentially binds to smaller species and/or potentially generates smaller species. There are several issues with this figure: (1) it is not clear whether fully cleaved protein was used. The strong absorbance at 280 nm could have come from partially cleaved protein, which contains Trp residues in the Trx portion. Cleavage of the fusion protein typically causes a mixture of cleaved and uncleaved proteins, which tend to co-aggregate and are very difficult to separate using the methods employed by the authors. There are no molecular weight controls that could give some insight into the potential amounts of fusion protein. (2) SERF1a is presented with a heterogeneous mixture of mostly pre-aggregated proteins. (3) A smaller species is generated, but it is argued that a larger aggregate forms. (4) Fraction 17 appears to have beta-sheet structure according to CD. This is confusing as a small aggregate is formed, which supposedly is held together by helical structure. Overall, Figure 5 remains weak.

Supplementary figure 4 was added to show that SERF1a does not bind to the final fibrillar aggregates. Immunogold labeling is informative, when it works and when it gives positive results. However, the absence of a signal in immunogold labeling is a negative result and does not necessary mean that SERF1a did not bind to fibrils. It is possible that the antibody cannot detect SERF1a bound to fibrils. The Western blot suggest that SERF1a is only in the supernatant. It would be more informative to also show an actual SDS gel directly.

Reviewer 3 Q3.

The ThT data are noisy, but if taken at face value, there is a very marginal decrease in the lag time for A β aggregation, even at 1:1 ratios. The amplitude change for alpha-synuclein is present, albeit also marginal. It should be noted, that ThT amplitude cannot directly be equated to amounts. This should be acknowledged in the text. This point also applies to the Httex1 data. In the case of Q49, only an amplitude change is observed whereas the actual rate constants (not rate) would likely be close to identical. The only difference would come from a single data point (~10h) that appears to be an outlier. This is an important point, as the overall premise is that SERF1a increases aggregation kinetics. At present, this is not convincing or, at least it does not relate to ThT positive aggregates.

Q5:

The binding studies remain confusing. It is not clear why the polyQ seems to inhibit binding (Fig.4, Htt-0) when the polyQ expansion seems to be helpful for binding in the prior figures. The most helical peptide fragment (Htt-1) also does not appear to bind. This makes it harder to understand how α -helical structure is thought to promote this interaction. I am also unconvinced with respect to the effect of Trx cleavage. The curves for 39Q in 2a and 2b are rather different. Furthermore, the work with recombinant protein might have a heterogeneous mixture making it harder to be quantitative.

Other points that should be addressed.

- It was cumbersome to understand which constructs were used. At a minimum this needs to be clear from the nomenclature. The Trx portion should be in the name that is used. It appears that a thioredoxin fusion protein was used for many experiments. The fusion partner is located at the N-terminus, making it extremely likely to interfere with N-terminal binders. This could severely alter the conclusions of the paper. Furthermore, Trx reduces aggregation propensity and most likely changes the aggregation mechanism. In other words, whatever is learned here may not apply to the actual Httex1 protein. This makes many aggregation experiments quite problematic. Additionally, Trx contains Trp residues that could contribute to the fluorescence experiments. It is also notoriously difficult to separate cleaved and uncleaved proteins through Ni columns.
- In many parts, it was not clear what the concentrations and conditions are. For example, I could not find the concentrations for the CD experiments. This would have been needed to understand the results. There are seem to be beta sheet contributions in the CD spectra of Figure 1. Could those be from Trx? Was this construct used here?

Reviewer #1 (Remarks to the Author):

The revised version of the manuscript “Amyloid modifier SERF1a interacts with PolyQ-expanded huntingtin-exon1 via helical interactions and exacerbates polyQ-induced toxicity” addresses the majority of points that I raised before. I appreciate the efforts to improve the study, with an important number of new experiments. Despite these efforts, I am not fully convinced about the highest resolution details of the mechanism based on the interpretation of the NMR and SAXS data.

- I am not convinced that the interpretation of the massive decrease of the SERF1a NMR intensities upon addition of HttEx1 is correct. It is true that the HttEx1 construct is large... but it is also true that it is highly flexible. Thus, I would not expect a systematic intensity decrease for all residues, including the most terminal ones. Maybe it would help if the intermediate titration point was shown, to see a dose dependent effect on the NMR intensities

Ans: We thank the reviewer for this comment. Here, we attached the intermediate titration points as suggested by the reviewer. Our data did show a dose-dependent effect on the NMR intensities (see figure below). We added this result in the revised manuscript in Supplementary Fig. 7 (See Page 9, Line 12).

- The SAXS data are also intriguing. I appreciate the efforts to measure new SEC-SAXS data, which should be described in the Methods section. The full SEC profile should be shown. It is unclear that for mixtures, the complex is captured taking into account the

dilution caused by the column and the transient nature of the interaction. It could be that the authors capture an unbound mixture of both peptides, which would explain the small R_g found for the mixture when compared with SERF1a alone.

I think that the authors should try to address these points and be less conclusive with respect to the NMR and SAXS experiments. The rest of the data seem solid and relevant.

Ans: We thank the reviewer for this comment. The description of the SEC-SAXS method is added to the revision (See Page 24, Line 19). The SEC-SAXS profiles were overlapped well; the correspondingly evolution of the radius of gyration R_g extracted are shown in Figure S12. The corresponding R_g values extracted from the SAXS profiles over the whole sample evolution of the mixture are nearly the same, indicating that there is only one single complex species in the mixture, rather than an average of the R_g sizes of the relatively smaller NT17 or Htt3 and large SERF1a species speculated by the reviewer.

Reviewer #2 (Remarks to the Author):

Response to reviewers 2 and 3.

The relevance mentioned by reviewer 2 was addressed.

Reviewer 3. Question 1.

The concerns regarding binding of SERF1a to beta sheet containing stages of Httex1 aggregation are not well addressed.

Figure 5 presents data showing that SERF1 preferentially binds to smaller species and/or potentially generates smaller species. There are several issues with this figure: (1) it is not clear whether fully cleaved protein was used. The strong absorbance at 280 nm could have come from partially cleaved protein, which contains Trp residues in the Trx portion. Cleavage of the fusion protein typically causes a mixture of cleaved and uncleaved proteins, which tend to co-aggregate and are very difficult to separate using the methods employed by the authors. There are no molecular weight controls that could give some insight into the potential amounts of fusion protein.

Ans: We are sorry for the confusion. The Httex1-49Q protein we used in Figure 5 was uncleaved protein with Trx tag, so there is no issue about a mixture of cleaved and uncleaved proteins. And in the revised manuscript we added molecular weight controls for the SEC in Supplementary Fig. 14.

(2) SERF1a is presented with a heterogeneous mixture of mostly pre-aggregated

proteins.

Ans: Based on our AUC SV data as shown in Figure 5a and zoom in Supplementary Fig. 13, there were mostly monomer with very few pre-aggregated proteins in the presence of SERF1a. Therefore, SERF1a was present a predominantly monomeric TrxHttex1-49Q.

(3) A smaller species is generated, but it is argued that a larger aggregate forms. (4) Fraction 17 appears to have beta-sheet structure according to CD. This is confusing as a small aggregate is formed, which supposedly is held together by helical structure. Overall, Figure 5 remains weak.

Ans. Since our data showed very few pre-aggregates were present, the concern of forming a larger aggregate does not exist. The result suggested that SERF1a transformed monomeric TrxHttex1-49Q to a unique β -sheet-rich conformation, which is more prone to aggregate.

Supplementary figure 4 was added to show that SERF1a does not bind to the final fibrillar aggregates. Immunogold labeling is informative, when it works and when it gives positive results. However, the absence of a signal in immunogold labeling is a negative result and does not necessary mean that SERF1a did not bind to fibrils. It is possible that the antibody cannot detect SERF1a bound to fibrils. The Western blot suggest that SERF1a is only in the supernatant. It would be more informative to also show an actual SDS gel directly.

Ans: We thank the reviewer for this suggestion. Here, we showed the actual SDS gel in the revised manuscript (see Supplementary Fig. 5b). The result was consistent with the western blot showing that SERF1a was only present in the supernatant.

Reviewer 3 Q3.

The ThT data are noisy, but if taken at face value, there is a very marginal decrease in the lag time for A β aggregation, even at 1:1 ratios. The amplitude change for alpha-synuclein is present, albeit also marginal. It should be noted, that ThT amplitude cannot directly be equated to amounts. This should be acknowledged in the text. This point also applies to the Httex1 data. In the case of Q49, only an amplitude change is observed whereas the actual rate constants (not rate) would likely be close to identical. The only difference would come from a single data point (~10h) that appears to be an outlier. This is an important point, as the overall premise is that SERF1a increases aggregation kinetics. At present, this is not convincing or, at least it does not relate to ThT positive aggregates.

Ans: We thank the reviewer for raising this point. For the note, we added in the revised manuscript that “However, it should be aware that the amplitude of ThT signal cannot directly be considered as the amounts of the fibrils (Page 5, Line 23)”. In the case of Q49, although there is only one data point at ~10 h showing larger difference, we don’t consider it as an outlier since other data including dot blot (Supplementary Fig. 3) and CD (Figure 1e) showed that Q49 aggregation was accelerated by SERF1a and reached the steady state more quickly. Our dot blot results (Supplementary Fig. 3) support this finding as the signals of A11 and OC antibodies significantly increased at 12 h of incubation in the presence of SERF1a. Also, the CD data (Figure 1e) showed a faster conformational transition in the presence of SERF1a. Besides, although ThT amplitude cannot directly be equated to fibril amounts, our filter-trap data (Fig. 1b) and TEM images (Fig. 1c) showed that SERF1a enhanced the fibril formation of Q49.

Q5:

The binding studies remain confusing. It is not clear why the polyQ seems to inhibit binding (Fig.4, Htt-0) when the polyQ expansion seems to be helpful for binding in the prior figures. The most helical peptide fragment (Htt-1) also does not appear to bind. This makes it harder to understand how α -helical structure is thought to promote this interaction. I am also unconvinced with respect to the effect of Trx cleavage. The curves for 39Q in 2a and 2b are rather different. Furthermore, the work with recombinant protein might have a heterogeneous mixture making it harder to be quantitative.

Ans: We thank the reviewer for this comment. Actually, we have discussed this in the discussion of our previous submission: “SERF1a did not bind to Htt-0, possibly due to the dimerization of Htt-0 and/or the short polyQ in Htt peptides. Similarly, Htt-1 formed pentamers and octamers that may hindered the binding to SERF1a (Supplementary Fig. 9). In fact, NT17 was shown to adopt different conformations when attached to short and long polyQ tract. Therefore, SERF1a and Httex1 interaction may be facilitated due to conformational changes of NT17 in long poly tract.” (Page 15, Line 15).

The Trx fusion tag was used to improve the solubility of polyQ proteins; however, we cannot totally rule out that there might be effect on interaction between SERF1a and polyQ. We have added this point in the discussion in the revised manuscript (Page 15, last Line). Although the curves for 39Q in 2a and 2b showed some differences in the binding affinity, the conclusion for whether it binds was not changed, suggesting the effect of Trx tag is marginal. Since we used TrxHttex1-polyQ in most of the experiments except for Figure 2b, there is no problem with having heterogeneous mixtures in our study.

Other points that should be addressed.

- It was cumbersome to understand which constructs were used. At a minimum this needs to be clear from the nomenclature. The Trx portion should be in the name that is used. It appears that a thioredoxin fusion protein was used for many experiments. The fusion partner is located at the N-terminus, making it extremely likely to interfere with N-terminal binders. This could severely alter the conclusions of the paper. Furthermore, Trx reduces aggregation propensity and most likely changes the aggregation mechanism. In other words, whatever is learned here may not apply to the actual Httex1 protein. This makes many aggregation experiments quite problematic. Additionally, Trx contains Trp residues that could contribute to the fluorescence experiments. It is also notoriously difficult to separate cleaved and uncleaved proteins through Ni columns.
- In many parts, it was not clear what the concentrations and conditions are. For example, I could not find the concentrations for the CD experiments. This would have been needed to understand the results. There are seem to be beta sheet contributions in the CD spectra of Figure 1. Could those be from Trx? Was this construct used here?

Ans: We are sorry for the confusion. In the revised manuscript, we clearly named Httex1 with Trx tag as TrxHttex1 and made the concentrations and conditions clearer in the text for the revision. We also acknowledged that the results we found in TrxHttex1 may not totally apply to the actual Httex1 protein in the discussion. However, by incubating thioredoxin with SERF1a, we found no effect of SERF1a on the secondary structure of thioredoxin alone. Therefore, we may exclude the SERF effect on thioredoxin. We added this result in the text as Supplementary Fig. 4.

REVIEWERS' COMMENTS:

Reviewer #1 (Remarks to the Author):

The authors have addressed all the points I raised. The manuscript can be published.

Reviewer #2 (Remarks to the Author):

The revised version is much improved. I was pleased to see that there is a cautionary statement in the Discussion stating that by not cleaving the Trx fusion protein not all conclusions may apply. I feel it would be important to add that Trx is generally used to slow down aggregation (that is why the tag is used for purification). It may also interfere with N-terminal binders. Both points should be stated so the reader is aware what the caveat means.